# GARMENTGPT: COMPOSITIONAL GARMENT PATTERN GENERATION VIA DISCRETE LATENT TOKENIZATION

**Fangsheng Weng**[1,*,§]    **Junhao Chen**[2,*]    **Xiang Li**[3]    **Jie Qin**[4]    **Hanzhong Guo**[5]
**Shaochun Hao**[‡]    **Xiaoguang Han**[4,6,7,†]

[1]ChimerAI    [2]Shenzhen International Graduate School, Tsinghua University
[3]The Hong Kong University of Science and Technology, Guangzhou
[4]School of Science and Engineering, The Chinese University of Hong Kong, Shenzhen
[5]School of Computer Science, The University of Hong Kong
[6]Shenzhen Future Network of Intelligence Institute
[7]Guangdong Provincial Key Laboratory of Future Networks of Intelligence,
The Chinese University of Hong Kong, Shenzhen

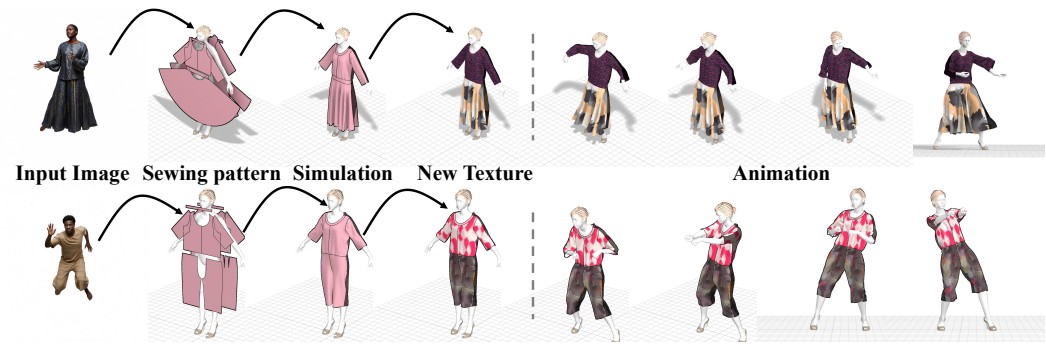

Figure 1: GarmentGPT generates sewing patterns from text and image inputs, maintaining robustness across arbitrary complex poses and garment textures.

## ABSTRACT

Apparel is a fundamental component of human appearance, making garment digitalization critical for digital human creation. However, sewing pattern creation traditionally relies on the intuition and extensive experience of skilled artisans. This manual bottleneck significantly hinders the scalability of digital garment creation. Existing generative approaches either operate as data replicators without intrinsic understanding of garment construction principles (e.g., diffusion models), or struggle with low-level regression of raw floating-point coordinates (e.g., Vision-Language Models). We present GarmentGPT, the first framework to operationalize latent space generation for sewing patterns. Our approach introduces a novel pipeline where a RVQ-VAE tokenizes continuous pattern boundary curves into discrete codebook indices. A fine-tuned Vision-Language Model then autoregressively predicts these discrete token sequences instead of regressing coordinates, enabling high-level compositional reasoning. This paradigm shift aligns generation with the knowledge-driven, symbolic reasoning capabilities of large language models. To address the data bottleneck for real-world applications, we develop a Data Curation Pipeline that synthesizes over one million photorealistic images paired with GarmentCode, and establish the Real-Garments Benchmark for comprehensive evaluation. Experiments demonstrate that GarmentGPT significantly outperforms existing methods on structured datasets (95.62% Panel Accuracy, 81.84% Stitch Accuracy), validating our discrete compositional paradigm's advantages. Code is available at https://github.com/ChimerAI-MMLab/Garment-GPT.

---

[†]Corresponding author    [*]Equal contribution    [§]Project Leader    [‡]Independent Researcher

# 1 INTRODUCTION

The emerging fields of digital humans, virtual reality, and e-commerce have spurred immense demand for high-fidelity digital apparel design. Sewing patterns serve as the foundational blueprint for garment creation, directly encoding geometric and topological information that determines the final 3D shape, fit, and style. However, the creation of sewing patterns is a notoriously complex, traditionally reliant on the intuition and extensive experience of skilled artisans. This manual bottleneck significantly hinders the scalability of digital garment creation, making the automated generation of sewing patterns from intuitive, multi-modal inputs a critical challenge with profound research and commercial value.

Early attempts ranged from rule-based systems to deep learning models reconstructing patterns from 3D scans (Korosteleva & Lee, 2022b) or images (Liu et al., 2023b). Recent approaches fall into two paradigms, both operating in "raw data space" without high-level reasoning. ***First***, generative models like diffusion (Liu et al., 2025) excel at learning distributions but lack intrinsic understanding of garment construction principles, functioning as data replicators disconnected from symbolic reasoning (Korosteleva & Lee, 2022a). ***Second***, Vision-Language Models (VLMs) (Bian et al., 2025; Nakayama et al., 2025) face a critical mismatch: powerful reasoning engines forced to regress raw floating-point coordinates, which is a low-level process where errors accumulate catastrophically.

Our work is inspired by a paradigm shift seen in other domains, most notably the success of Latent Diffusion Models (Rombach et al., 2022). Instead of operating directly in the high-dimensional, raw data space, we first map the problem into a more compact, semantically rich latent space. The key insight is that sewing patterns, like other structured geometric data (Chen et al., 2024; Yang et al., 2025; Song et al., 2025b), possess an inherent discrete nature that can be effectively captured through vector quantization. Several discretization (Agustsson et al., 2017; Yang et al., 2023; Guo et al., 2025b)approaches have been explored in related domains by providing learned discrete representations to improve reconstruction quality.

We present **GarmentGPT**, the first framework operationalizing latent space generation for sewing patterns. A **Residual Vector Quantizer VAE (RVQ-VAE)** (Zeghidour et al., 2021) tokenizes continuous pattern curves into discrete codebook indices representing meaningful components (panels, curves, connections) combinable according to tailoring principles. A fine-tuned Vision-Language Model then autoregressively predicts these token sequences instead of regressing coordinates, enabling high-level compositional reasoning. This paradigm shift aligns generation with the knowledge-driven, symbolic reasoning capabilities of large language models.

Meanwhile, a major bottleneck for practical garment generation applications is the lack of large-scale datasets pairing real-world photographs with their corresponding precise sewing patterns. This data gap prevents training models for practical use cases involving real photo inputs. To address this critical challenge, we propose a novel data curation pipeline that leverages automated pose variation and appearance editing. The pipeline synthesizes from structured GarmentCode through simulation and rendering a dataset of over one million paired real human images and garment patterns with diverse poses, fabric textures, and garment fits—**RealGarment-1M (RG-1M)**. We also introduce **RealGarmentTexture-164K (RGT-164K)**, a dataset containing 164K unique flattened garment textures, enabling models to learn from large-scale real-world-like inputs for the first time. To better evaluate real image-to-pattern generation, we establish **RealGarment-Bench (RG-Bench)**, a comprehensive benchmark introducing extensive variations in human poses and texture patterns under identical garment patterns, enabling evaluation in a manner closer to human usage scenarios.

In summary, our main contributions are as follows:

- We introduce **GarmentGPT**, a novel generative pipeline built upon **Residual Vector Quantization (RVQ)** that transforms pattern generation into a knowledge-driven, compositional task in semantic latent space, enabling structurally accurate and editable patterns from multi-modal inputs.
- We propose a novel **Data Curation Pipeline** and introduce two foundational datasets synthesized through automated pose variation and appearance editing: **RealGarment-1M** (1M+ photorealistic image-pattern pairs with diverse poses, textures, and fits) and **RealGarmentTexture-164K** (164K unique flattened textures), providing the first large-scale resources for real-world garment generation.
- We establish **RealGarment-Bench**, the first comprehensive benchmark for evaluating garment generation from real human photos, assessing both visual fidelity and structural correctness.

- Extensive experiments demonstrate our approach significantly outperforms existing methods across all metrics, achieving state-of-the-art performance for garment generation from real photos.

## 2 RELATED WORK

### 2.1 DIGITAL GARMENT DESIGN

Digital garment design spans multiple paradigms. **Image-Based Methods** such as text-to-image (Baldrati et al., 2023; Zhang et al., 2022; 2024b) and image-to-image translation (Chong et al., 2024; 2025; Zhang et al., 2025b; Chen et al., 2023a) produce photorealistic visualizations but lack structural information for editing or manufacturing. **3D Synthesis Methods**, including text/image-to-3D (Poole et al., 2022; Chen et al., 2025b; Long et al., 2024; Miao et al., 2026; Zhao et al., 2025; Chen et al., 2026; Zhang et al., 2024c), video generation (Wang et al., 2024d; Karras et al., 2023; Hu, 2024; Tu et al., 2025; Chen et al., 2025a; Ma et al., 2025), and character animation (Xu et al., 2020; Zhang et al., 2025a; Song et al., 2025a; Sun et al., 2025; Guo et al., 2025c), yield mesh or point cloud outputs that cannot encode 2D panel layouts, stitching, or parametric constraints needed for production. **Structured Pattern Representations** (Korosteleva & Sorkine-Hornung, 2023; Liu et al., 2023b) instead encode garments as 2D panels, curves, and stitching relationships, enabling direct use in manufacturing. Our work advances this paradigm via a novel discrete tokenization scheme that preserves structural integrity.

### 2.2 STRUCTURED DATA GENERATION

Fine-tuning pre-trained foundation models is the de facto standard for domain adaptation. In NLP, parameter-efficient techniques such as LoRA (Hu et al., 2022) and prompt tuning (Lester et al., 2021; Li & Liang, 2021) yield domain experts like BioGPT (Luo et al., 2022) and LegalBERT (Chalkidis et al., 2020), while efficient task-specific architectures (Chen et al., 2023b) and comprehensive benchmarks (Zhang et al., 2023; Chen et al., 2025c) further drive progress. Vision-Language Models (VLMs) (Radford et al., 2021; Liu et al., 2023a; Wang et al., 2024a) have shown strong multimodal understanding across diverse tasks (Ye et al., 2024; Guo et al., 2025a) and have been applied to structured data generation, including 3D meshes (Wang et al., 2024e; Chen et al., 2024; Tang et al., 2024), skeletons (Song et al., 2025b), vector graphics (Yang et al., 2025; Liu et al., 2024b), and CAD (Alrashedy et al., 2024). However, garment patterns pose unique challenges: VLM token limits conflict with the need for precise geometric constraints and explicit inter-panel topology. Existing codebook methods (Yu et al., 2021; Ji et al., 2024) do not address these manufacturing requirements; we propose a tokenization scheme specifically tailored to sewing pattern geometry and topology.

### 2.3 SEWING PATTERN GENERATION

Large-scale datasets like CLOTH3D (Bertiche et al., 2020), GarmentCode (Korosteleva et al., 2024) and Dress Code (Morelli et al., 2022) have enabled data-driven pattern generation through three main approaches: **(1) Optimization-Based:** Sensitive Couture (Umetani et al., 2011) pioneered bidirectional 2D-3D editing, while FoldSketch (Li et al., 2018) enabled pleat design via 3D sketches. These methods rely on complex physical optimizations that limit scalability. **(2) Autoregressive:** GarmentCode (Korosteleva & Sorkine-Hornung, 2023) uses hierarchical parametric components, while DressCode (Morelli et al., 2022), AIpparel(Nakayama et al., 2025), SewFormer(Liu et al., 2023b),and ChatGarment (Bian et al., 2025) employ Transformers for sequential generation. Despite capturing fine details, they suffer from slow inference and error propagation. **(3) Feed-Forward:** StableGarment (Wang et al., 2024c) and SewingLDM (Liu et al., 2025) use diffusion models for parallel generation, offering efficiency but struggling with precise geometric constraints. This precision-efficiency trade-off motivates our approach.

## 3 METHOD

### 3.1 OVERVIEW

Our garment pattern generation methodology, as shown in Fig. 2, comprises two fundamental components: a quantization module that encodes edge and panel positional information, and a

sequence generation and editing module based on generative Vision-Language Models (VLMs) capable of comprehending multimodal inputs and generating or modifying encoded sequences.

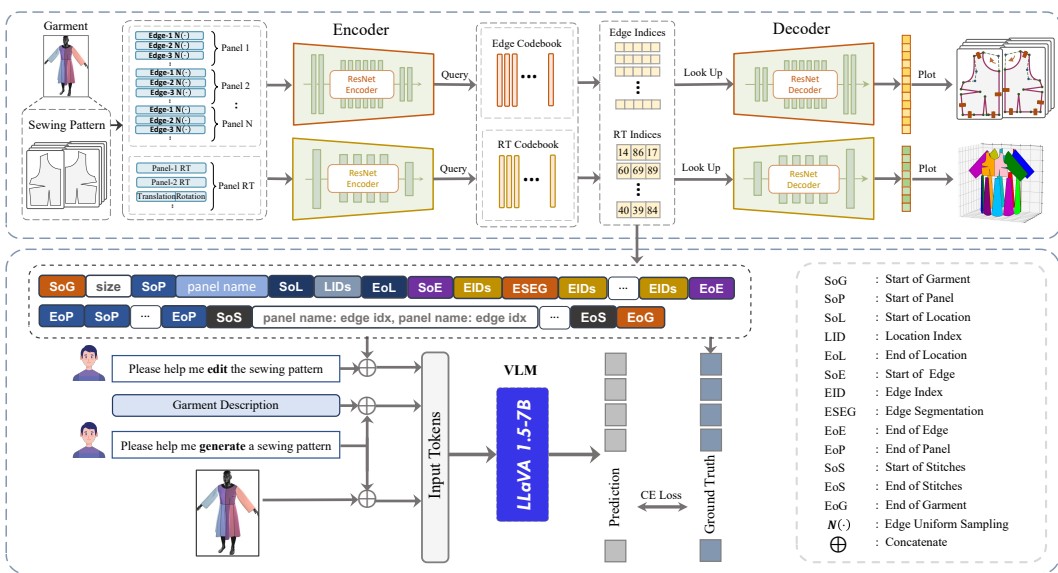

Figure 2: Overview of the GarmentGPT framework. Garment patterns are encoded into discrete token sequences via residual vector quantization with separate codebooks for edge geometry and panel positioning, then processed by a Vision-Language Model for multimodal generation and editing.

We encode edges into discrete index representations through residual vector quantization, replacing traditional continuous numerical representations while simultaneously generating corresponding codebooks for reconstruction. For each panel's translation and rotation parameters in SMPL A-pose (Loper et al., 2023) configuration, we employ an identical encoding-decoding workflow and allocate a separate, non-shared codebook distinct from edge encoding.

For complete garments, the encoded edge indices, panel translation and rotation parameter indices, and inter-edge stitching relationships are systematically combined through a carefully designed tokenization strategy into sequence data processable by VLMs, termed garment sequences. VLMs can perform multimodal comprehension based on images, text, or existing garment sequences, subsequently generating or editing garment sequences. The generated sequences are then deconstructed through pattern matching methodologies, querying corresponding code tables for decoding, ultimately outputting each garment's dimensions, each panel's translation and rotation parameters, and the positional and stitching relationships of each edge.

## 3.2 VLM BACKBONE AND TRAINING STRATEGY

We employ Vision-Language Models (VLMs), including LLaVA and QwenVL, as our foundational architecture for experiments. VLMs convert garment images into structured topological token sequences. These models excel at mapping pixel-level visual features to semantic concepts, making them ideal for perceiving images and generating structured outputs. We adapt VLMs to garment topological modeling through three key enhancements: **(1) Vocabulary Extension:** We add specialized topological tokens and codebook indices (0-1023) as learnable embeddings to the LLM vocabulary. **(2) Multimodal Input Construction:** For generation tasks, we use image-text pairs; for editing tasks, we use sequence-text pairs with editing instructions. Visual and textual inputs are processed and fused via projection layers. **(3) Training Paradigm:** We fine-tune the VLM to autoregressively predict topological token sequences from multimodal inputs, minimizing cross-entropy loss against ground-truth sequences.

Figure 3: **Hierarchical tokenization of garment patterns.** A Tank Top garment is converted into a structured token sequence using specialized tokens to encode garment structure, panels, positional parameters, edges, and stitching relationships. Detailed encoding process is described in Section 3.4.

## 3.3 QUANTIZATION OF SEWING PATTERNS

Sewing patterns consist of multiple panels with unique identifiers, spatial positioning (translation and rotation in A-pose), and contours formed by connected edges. Edges are classified into four geometric types: **straight lines**, **quadratic/cubic Bézier curves**, and **circular arcs**, each with specific parameters (vertices, control points, radius, etc.).

Our quantization framework employs three modules—encoder, codebook, and decoder—to achieve discrete representation and reconstruction of pattern information. To maintain representational purity, edges and RT (rotation-translation) parameters use separate encoder-decoder pairs and codebooks. Refer to the left panel of Fig. 3 for an example.

For edge encoding, we uniformly sample each edge into N points and process them through a lightweight ResNet-based encoder. RT parameters are concatenated into a single vector and encoded using a smaller ResNet architecture. Through Residual Quantization, continuous latent vectors are mapped to discrete indices via hierarchical codebooks, where vectors exhibit residual relationships. The number of residual levels controls the balance between compression efficiency and reconstruction quality.

During decoding, the system retrieves quantized vectors by querying indices and reconstructs parameters through symmetric decoders. Edge decoders restore endpoints and type-specific attributes (radius for arcs, control points for Bézier curves via lossless quadratic-to-cubic conversion). RT parameters directly regress to continuous values, enabling high-fidelity reconstruction of complete pattern structures.

For more details, please refer to Appendix Sec. C

## 3.4 TOKENIZATION

A pivotal aspect of fine-tuning VLMs lies in effectively representing the geometric topological structure of garment patterns as token sequences. To this end, we have designed a specialized garment pattern topological representation framework that introduces a set of special tokens $\mathcal{T}$ to encode different hierarchical levels and relationships within garment structures:

$$\mathcal{T} = \big\{ \langle\text{SoG}\rangle, \langle\text{EoG}\rangle, \langle\text{SoP}\rangle, \langle\text{EoP}\rangle, \langle\text{SoL}\rangle, \langle\text{EoL}\rangle, \langle\text{SoE}\rangle, \langle\text{EoE}\rangle, \langle\text{SoS}\rangle, \langle\text{EoS}\rangle, \langle\text{ESEG}\rangle \big\} \quad (1)$$

Each token pair marks the beginning and end of specific semantic units, corresponding to the overall garment, panels, positional parameters, edges, and stitching relationships, respectively; additionally, the $\langle\text{ESEG}\rangle$ token serves as a delimiter to distinguish index sequences of different edges within the same panel.

**Hierarchical Structure.** This representation methodology employs a hierarchical organizational structure, as illustrated in Fig. 3. The complete garment commences with $\langle\text{SoG}\rangle$, followed by dimensional information, all panel data, and stitching relationships, ultimately concluding with $\langle\text{EoG}\rangle$. Each panel's information initiates with $\langle\text{SoP}\rangle$, encompassing its identifier, positional parameters, and edge sequences arranged according to drawing order, terminating with $\langle\text{EoP}\rangle$. Panel positional information is denoted by $\langle\text{SoL}\rangle$, succeeded by index values organized in $[a, b, c]$ format, and concludes with $\langle\text{EoL}\rangle$. Edge information within panels begins with $\langle\text{SoE}\rangle$, followed by corresponding index sequences in the codebook, with $\langle\text{ESEG}\rangle$ separating indices of different edges, and the entire

edge information terminates with $\langle\text{EoE}\rangle$. Stitching information is encapsulated between $\langle\text{SoS}\rangle$ and $\langle\text{EoS}\rangle$, specifically describing edge pair relationships requiring stitching.

**Concrete Example.** Fig. 3 illustrates a tokenized representation of a Tank Top garment. This garment measures 93 centimeters in dimension (defined as the maximum absolute value of all panel vertex coordinates), comprising 4 panels (right_btorso, left_btorso, left_ftorso, and right_ftorso), with each panel consisting of 13 edges. The garment exhibits various stitching relationships, including edge connections within the same panel (e.g., the 11th and 12th edges of right_ftorso) and between different panels (e.g., the 4th edge of right_btorso and the 4th edge of left_btorso, shown in identical colors). In this example, we use 5 residual indices for edge encoding and 3 for positional parameter encoding, with each index ranging from 0 to 1023 (codebook size of 1024). Through this tokenization methodology, the complete garment is transformed into the discrete token sequence shown in the figure.

## 3.5 Loss Function

We design separate loss functions for the quantizer and VLM components.

**Quantizer Loss.** For the quantizer component, the overall loss comprises multiple weighted sub-losses to comprehensively assess encoding and reconstruction quality:

$$\mathcal{L}_{\text{quant}} = \lambda_{\text{cls}}\mathcal{L}_{\text{cls}} + \lambda_{\text{vertex}}\mathcal{L}_{\text{vertex}} + \lambda_{\text{control}}\mathcal{L}_{\text{control}} + \lambda_{\text{commit}}\mathcal{L}_{\text{commit}} \tag{2}$$

where $\lambda_{\text{cls}}$, $\lambda_{\text{vertex}}$, $\lambda_{\text{control}}$, and $\lambda_{\text{commit}}$ represent the weighting coefficients for each loss component. Each component is defined as follows:

**Classification Loss ($\mathcal{L}_{\text{cls}}$):** This loss accurately predicts the geometric type of each edge (straight lines, Bézier curves, or circular arcs) using standard cross-entropy loss.

**Vertex Loss ($\mathcal{L}_{\text{vertex}}$):** This loss constrains reconstruction precision of curve endpoints by computing the $L_2$ distance between predicted and ground-truth endpoints:

$$\mathcal{L}_{\text{vertex}} = \|\mathbf{v}_{\text{pred}} - \mathbf{v}_{\text{gt}}\|_2^2 \tag{3}$$

where $\mathbf{v}_{\text{pred}}$ and $\mathbf{v}_{\text{gt}}$ denote the predicted and ground-truth vertex coordinates, respectively.

**Control Point Loss ($\mathcal{L}_{\text{control}}$):** This loss is computed differentially for distinct curve types. For straight lines, we use two trisection points as proxy control points. For Bézier curves, we calculate the $L_2$ distance between predicted and ground-truth control points. For circular arcs, the loss comprises three components: (1) $L_2$ distance for the radius parameter, (2) binary cross-entropy loss for the major/minor arc indicator, and (3) binary cross-entropy loss for the drawing direction:

$$\mathcal{L}_{\text{control}}^{\text{arc}} = \|r_{\text{pred}} - r_{\text{gt}}\|_2^2 + \text{BCE}(d_{\text{pred}}, d_{\text{gt}}) + \text{BCE}(a_{\text{pred}}, a_{\text{gt}}) \tag{4}$$

where $r$, $d$, and $a$ represent the radius, drawing direction, and major/minor arc indicator, respectively.

**Commitment Loss ($\mathcal{L}_{\text{commit}}$):** This loss enhances correspondence relationships between encoding vectors and codebook entries, driving continuous codebook updates during training:

$$\mathcal{L}_{\text{commit}} = \beta\|z_e(\mathbf{x}) - \text{sg}(z_q)\|_2^2 \tag{5}$$

where $z_e(\mathbf{x})$ denotes the encoder output (continuous feature vector), $z_q$ represents the quantized codebook vector obtained through nearest-neighbor search, $\text{sg}(\cdot)$ denotes the stop-gradient operation, and $\beta$ is a hyperparameter controlling the commitment loss weight.

**VLM Loss.** For the VLM component, we employ the standard cross-entropy loss function to maximize the model's conditional likelihood probability for target sequences. This loss operates through autoregressive methodology, predicting the probability distribution of the next token based on previously generated tokens:

$$\mathcal{L}_{\text{VLM}} = -\frac{1}{N}\sum_{i=1}^{N}\log P\left(\text{token}_i \mid \text{token}_{<i}, \mathcal{C}\right) \tag{6}$$

where $\mathcal{C}$ represents the conditional input context (such as images, text, or structured sequences), and $N$ denotes the total sequence length. This loss function effectively drives the model to learn complex mapping relationships between multimodal inputs and structured garment sequences, constituting the core mechanism for achieving sequence generation and editing functionality.

## 4    DATA CURATION PIPELINE

Existing Image-to-GarmentCode methods trained on SMPL-rendered datasets exhibit poor generalization on real photos. While substantial GarmentCode data exists, obtaining corresponding real human images at scale remains the primary bottleneck for training image-conditioned generation models.

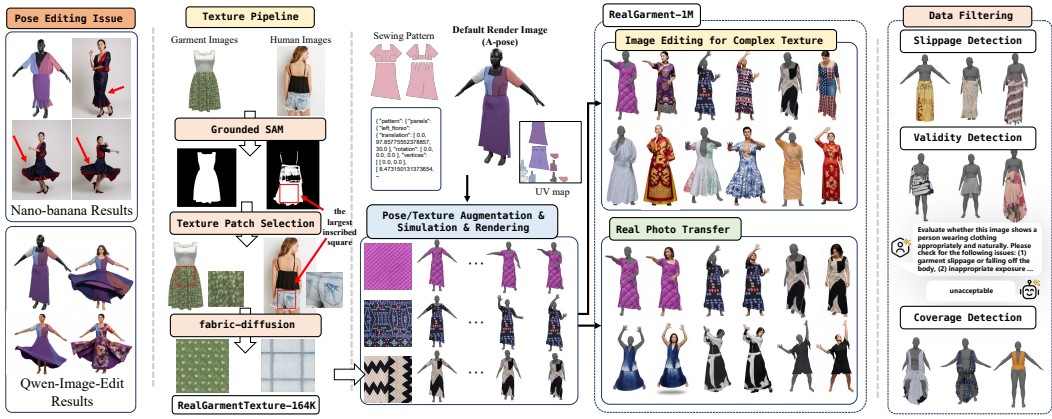

Figure 4: **Data curation pipeline overview.** Our pipeline generates photorealistic human images from GarmentCode through texture extraction, motion-aware simulation, photorealistic translation, and quality filtering.

We propose an automatic data curation pipeline that generates photorealistic human images from GarmentCode while preserving garment structure consistency (Fig. 4). Fig. 5 illustrates representative examples from our dataset, showcasing the smooth interpolation process from A-pose to diverse poses and the resulting photorealistic human images. The pipeline consists of four stages: (1) **Texture extraction and augmentation**: We build a texture extraction pipeline using Grounded-SAM (Ren et al., 2024) and FabricDiffusion (Zhang et al., 2024a) to process existing datasets (Liu et al., 2016; Huang et al., 2020; Cimpoi et al., 2014; Shakir & Topal, 2022; Liu et al., 2024a), yielding RGT-164K with 164K unique texture maps. (2) **Motion-aware simulation**: We extract SMPL poses from AMASS (Mahmood et al., 2019) and render clothed human videos with varying poses and textures using GarmentCode (Korosteleva & Sorkine-Hornung, 2023) and Contourcraft (Grigorev et al., 2024). (3) **Photorealistic translation**: Selected keyframes are converted to photorealistic images via Qwen-image-edit (Wu et al., 2025a), ensuring garment structure consistency through physics-based simulation. (4) **Quality filtering**: We implement multi-stage filtering including garment detachment detection, visual plausibility assessment via VLM, and coverage verification, improving alignment acceptability from 64.3% to 99.6%.

This pipeline produces RealGarment-1M (RG-1M), containing over one million photorealistic human image-GarmentCode pairs with diverse textures, poses, and appearances. **See Appendix Sec. A for implementation details and dataset statistics.**

## 5    EXPERIMENTS

### 5.1    EXPERIMENTAL SETUP

Our RVQ-VAE encoder consists of an 8-layer ResNet backbone, which maps continuous boundary curves (lines, circular arcs, and cubic Bézier curves), each sampled into 32 points, into discrete tokens. We use a codebook of size 1024. The codec is trained on a dataset of approximately 35 million curve segments. For the generative component, we fine-tune a Large Vision-Language Model (LLaVA-1.5 7B and Qwen-2.5-VL) on our curated dataset. The model is trained for 18 epochs with a sequence length of 4096 for LLaVA-1.5 7B and 16384 for Qwen-2.5-VL.

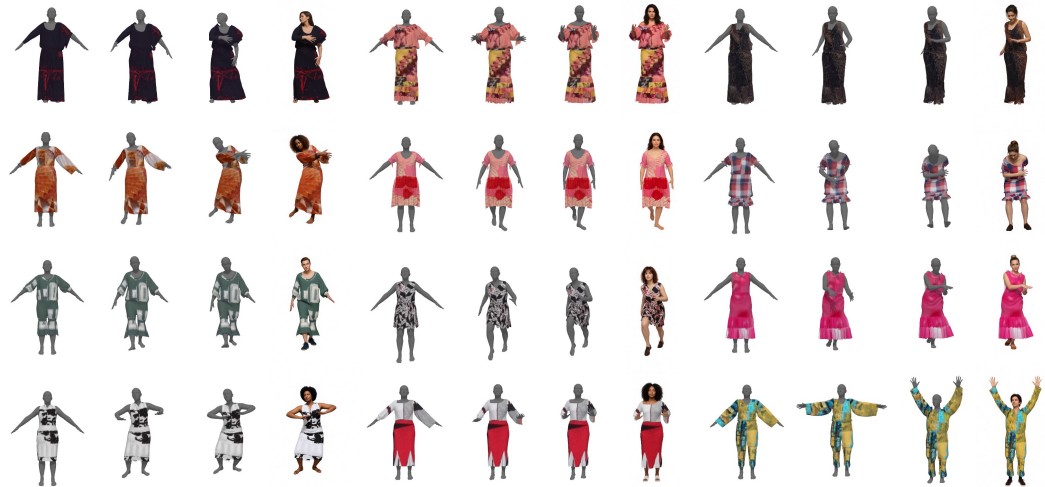

Figure 5: We use A-pose as the first frame and the target pose as the last frame for easing interpolation. For each case, the leftmost real human photo is converted from its adjacent rendering.

## 5.2 BASELINES AND METRICS

We use two primary evaluation settings: **(1) GarmentCode Dataset.** A large-scale structured dataset containing GarmentCode representations paired with corresponding rendered images and textual descriptions. This is used for direct comparison with baselines such as AIpparel and ChatGarment. **(2) Real-Garments Benchmark.** A novel benchmark created using our **Data Curation Pipeline**, featuring over 2,000 photorealistic images of virtual humans in diverse poses, paired with their ground-truth GarmentCode. This is used to evaluate performance on real human photo inputs.

We evaluate state-of-the-art GarmentCode generation models, including SewFormer (Liu et al., 2023b), ChatGarment (Bian et al., 2025), and AIpparel (Nakayama et al., 2025).

We evaluate performance using a comprehensive set of metrics. **Structural Accuracy** is measured by *Panel Accuracy*, *Edge Accuracy*, and *Stitch Accuracy*. **Geometric Error** is calculated as the L2 distance for *Panel Vertices*, *Panel Midpoints*, *Rotation*, and *Translation*. For our **Real-Garments Benchmark**, we additionally report **Practical Precision** using *Vertex Accuracy @ X mm*, which measures the percentage of vertices within a specified millimeter tolerance of the ground truth. These metrics are consistent with the AIpparel (Nakayama et al., 2025) setting.

## 5.3 EVALUATION ON GARMENTCODE DATASET

In this section, we evaluate GarmentGPT's core performance on the structured GarmentCode dataset and compare it against the state-of-the-art regression-based model, AIpparel.

**Superiority over Continuous Regression.** As summarized in Tab. 1, GarmentGPT significantly outperforms AIpparel in the multi-modal (Image+Text) setting. We achieve a **95.62%** Panel Accuracy compared to AIpparel's 78.92% (+16.7%) and an **81.84%** Stitch Accuracy versus their 56.57% (+25.3%). This stark difference highlights the fundamental advantage of our discrete, compositional paradigm. By reframing generation as a token selection task, GarmentGPT leverages the VLM's symbolic reasoning strengths, avoiding the error accumulation inherent in regressing thousands of continuous coordinates.

**Effectiveness of Multi-modality and Editing.** Our ablations confirm that GarmentGPT effectively fuses multi-modal inputs. The combination of image and text yields the best results, improving Panel Accuracy from 93.53% (image-only) to 95.62%. Furthermore, the model's exceptional performance on the editing task (e.g., **92.95%** Stitch Accuracy) demonstrates that our discrete representation provides a semantically meaningful and robust structure for granular modifications.

Table 1: Main results on the structured **GarmentCode** dataset.

| Method | Setting / Backbone | Panel Acc. ↑ | Edge Acc. ↑ | Stitch Acc. ↑ | Vertices L2 ↓ | Midpoint L2 ↓ | Rotation L2 ↓ | Translation L2 ↓ |
|---|---|---|---|---|---|---|---|---|
| **Baselines** | ChatGarment | 60.22% | 42.12% | 49.21% | 30.15 | 1910.42 | 10.51 | 10.03 |
| | AIpparel | 78.92% | 74.31% | 56.57% | 25.55 | 793.39 | 3.87 | 5.22 |
| **GarmentGPT (LLaVA-1.5)** | Text-only | 64.03% | 76.71% | 53.16% | 48.19 | 1507.18 | 0.84 | 8.80 |
| | Image-only | 93.53% | 89.75% | 80.98% | 17.33 | 543.86 | 0.56 | 2.93 |
| | Image+Text | **95.62%** | **90.48%** | **81.84%** | 18.43 | 577.95 | 0.59 | 3.05 |
| | Editing | 93.80% | 94.62% | 92.95% | 11.07 | 353.81 | 0.97 | 2.93 |
| **Backbone Ablation** | Qwen-3B | 85.56% | 81.32% | 69.13% | 733.22 | 23617.66 | 0.77 | 5.09 |
| | Qwen-7B | 90.31% | 85.79% | 74.77% | 46.99 | 1464.72 | 0.67 | 4.13 |
| | Qwen-32B | **91.05%** | 85.15% | **75.88%** | 51.20 | **1390.15** | 0.72 | **3.98** |

## 5.4 EVALUATION ON THE REAL-GARMENTS BENCHMARK

We evaluate generalization to real-world scenarios using our **Real-Garments Benchmark**, which features over 2,000 photorealistic images with diverse poses and appearances. This presents a significantly harder task than the clean A-pose renderings in GarmentCode. All methods are trained on our curated real photo dataset for fair comparison.

As shown in Tab. 2, all methods experience performance degradation, validating the benchmark's difficulty. However, **GarmentGPT maintains clear superiority**: while baselines suffer severe drops (ChatGarment: 25.34%, AIpparel: 38.76% Panel Accuracy), our method retains approximately 95% of its performance, achieving **90.84% Panel Accuracy**, a 2.3× improvement over the best baseline. This demonstrates that our discrete tokenization learns robust, pose-invariant representations, establishing GarmentGPT as the first practical solution for real-world garment digitization.

Table 2: Performance comparison on the challenging **Real-Garments Benchmark**.

| Method | Setting | Panel Acc. ↑ | Edge Acc. ↑ | Stitch Acc. ↑ | Vertices L2 ↓ | Midpoint L2 ↓ | Rotation L2 ↓ | Translation L2 ↓ |
|---|---|---|---|---|---|---|---|---|
| **Baselines** | ChatGarment | 25.34% | 17.82% | 18.45% | 71.28 | 4523.67 | 24.35 | 26.18 |
| | AIpparel | 38.76% | 41.25% | 27.34% | 58.92 | 1842.76 | 9.47 | 12.85 |
| **GarmentGPT (LLaVA-1.5)** | Image-only | 88.67% | 84.28% | 76.34% | 19.45 | 612.87 | 0.63 | 3.28 |
| | Image+Text | **90.84%** | **85.92%** | **77.56%** | **20.67** | **648.23** | **0.66** | **3.42** |

## 5.5 QUALITATIVE RESULTS

Fig. 6 presents qualitative comparisons between SewFormer, ChatGarment, AIpparel, and GarmentGPT on real human photo inputs. While baseline methods produce incomplete or distorted patterns, GarmentGPT consistently generates accurate, well-structured results that closely match the ground truth. Additional results on the Real-Garments Benchmark (Fig. 9, Fig. 10, Fig. 11) further demonstrate our method's robustness across diverse poses, garment types, and textures.

Additionally, Fig. 1 demonstrates potential applications of GarmentGPT-generated patterns in physical simulation, texture editing, and animation. Our method enables accurate garment pattern reconstruction from arbitrary real human photos, facilitating downstream workflows in garment design and manufacturing.

## 5.6 ABLATION STUDY ON RVQ-VAE CODEC

We validate the core of our framework by ablating the number of residual quantizers (Q) in RVQ-VAE. As shown in Tab. 3, increasing Q from 1 to 8 drastically improves reconstruction quality: total loss drops from 3.72 to 0.08, vertex loss decreases from $5.9 \times 10^{-3}$ to $0.05 \times 10^{-3}$, and curve accuracy reaches 99.8%. This confirms that Q=8 enables reliable pattern tokenization.

Table 3: Ablation study on the number of residual quantizers in our RVQ-VAE codec.

| Metric | Q=1 | Q=3 | Q=5 | Q=8 |
|---|---|---|---|---|
| Total Loss ↓ | 3.72 | 0.36 | 0.15 | **0.08** |
| Vertex Loss ↓ ($\times 10^{-3}$) | 5.9 | 0.39 | 0.14 | **0.05** |
| Curve Acc. ↑ (%) | 93.3 | 98.9 | 99.5 | **99.8** |

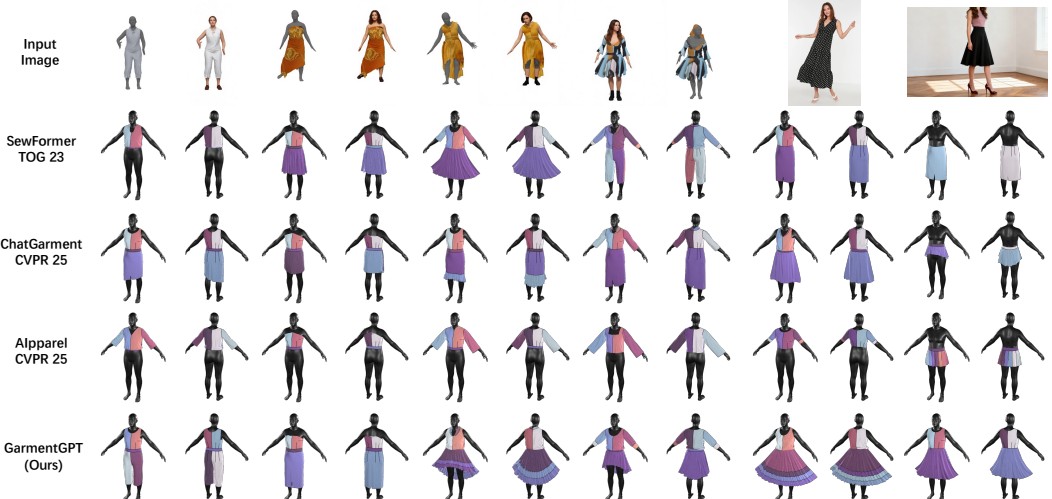

Figure 6: Each case shows the front and back renderings of garment code generated by different methods. The input images in the first row are real human photos. The left four cases use real human photos converted from SMPL renderings, while the right two cases use human photos from the Internet.

# 6 CONCLUSION

GarmentGPT introduces discrete tokenization via RVQ-VAE for sewing pattern generation, achieving 95.62% Panel Accuracy on structured data and 90.84% on real photos. We contribute RealGarment-1M (1M+ photorealistic pairs), RealGarmentTexture-164K, and Real-Garments Benchmark, enabling manufacturing-ready pattern generation from casual photos. This democratizes garment design beyond expert artisans.

ACKNOWLEDGEMENT

The work was supported in part by NSFC with Grant No. 62293482, the Basic Research Project No. HZQB-KCZYZ-2021067 of Hetao Shenzhen-HK S&T Cooperation Zone, by Guangdong Provincial Outstanding Youth Fund with No. 2023B1515020055, the Shenzhen Outstanding Talents Training Fund 202002, the Guangdong Research Projects No. 2017ZT07X152 and No. 2019CX01X104, the Guangdong Provincial Key Laboratory of Future Networks of Intelligence (Grant No. 2022B1212010001), and the Shenzhen Key Laboratory of Big Data and Artificial Intelligence (Grant No. SYSPG20241211173853027), the Guangdong Province Radio Science Data Center.

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

# A    DATA CURATION PIPELINE

Existing Image-to-GarmentCode methods trained on SMPL-rendered datasets exhibit poor generalization on real photos. While substantial GarmentCode data exists, obtaining corresponding real human images at scale remains the primary bottleneck for training image-conditioned GarmentCode generation models.

## A.1    PIPELINE OVERVIEW

To address this challenge, we propose an automatic data curation pipeline that generates **photorealistic human images** from GarmentCode while preserving garment structure consistency. Our pipeline consists of four main stages: (1) texture extraction and augmentation, (2) Simulation and rendering with variations in human pose, garment structure, and texture, (3) image-to-image translation for photorealism, and (4) quality filtering.

**Challenge: Pose-Induced Structure Drift.** Direct pose editing using existing text-conditioned (Wu et al., 2025a; Google, 2025; Wang et al., 2025a; Hurst et al., 2024; Wu et al., 2025b; Labs et al., 2025; Xie et al., 2025) or pose-conditioned (Tu et al., 2025; Wang et al., 2025b; Xu et al., 2025; Wang et al., 2024b;f; Hu, 2024; Sun et al., 2024; Chong et al., 2024; Choi et al., 2024; Kim et al., 2024; Fang et al., 2024; Velioglu et al., 2024; Chang et al., 2023; Morelli et al., 2023) image editing models alters garment structure. For example, as shown in Fig. 4 (left), a fitted cheongsam transforms into a flowing skirt after pose conversion. To ensure structure consistency, we render SMPL-based clothed human videos by varying human motion poses and garment textures through physics-based simulation (Grigorev et al., 2024), then convert selected keyframes to photorealistic images.

**Texture Extraction and Augmentation.** We built a texture extraction pipeline using Grounded-SAM (Ren et al., 2024) and FabricDiffusion (Zhang et al., 2024a) to process existing datasets, including DeepFashion (Liu et al., 2016), DeepFashion-MultiModal (Huang et al., 2020), DTD (Cimpoi et al., 2014), TFD (Shakir & Topal, 2022), and FID (Liu et al., 2024a). We use "clothing" as the text prompt to segment the garment region and obtain the corresponding mask, from which we select the largest inscribed square. As shown in the "Texture Patch Selection" step in Fig. 4, selecting the largest inscribed square within the mask region filters out segmentation failures (e.g., the segmentation mask of the black top fails in the Human Images input). We then employ FabricDiffusion (Zhang et al., 2024a) to extract flattened texture maps from clothed human and garment images. This process yields the RGT-164K dataset, containing 164K unique texture maps (see Tab. 4 for details).

**Motion Augmentation and Simulated Rendering.** For each garment pattern in GarmentCode, we perform data augmentation by varying garment textures and human poses. We extract random SMPL poses from AMASS (Mahmood et al., 2019) and generate smooth motions from the standard A-pose to the random poses through cosine interpolation. This ensures natural pose transitions while maintaining garment structure and preventing penetration artifacts during simulation. Using GarmentCode (Korosteleva & Sorkine-Hornung, 2023) and Contourcraft (Grigorev et al., 2024), we render SMPL-based clothed human videos with the above variations in a simulation environment. Each video starts from the A-pose as the first frame and ends at the selected random pose frame. We select the first frame (A-pose), middle frames, and the final frame (the pose sampled from AMASS) for the next stage.

**Image-to-Image Translation for Photorealism.** We employ Qwen-image-edit (Wu et al., 2025a) to convert rendered SMPL-based images into photorealistic human images. This translation step bridges the sim-to-real gap preserving garment structure, as the underlying 3D simulation **ensures consistent garment topology** across pose changes. The text prompt used is shown in Fig. 7.

```
Transform the SMPL model in this image into a photorealistic
human, while keeping the body pose, full-body clothing style,
texture, cut, size, and dimensions unchanged.
```

Figure 7: Text prompt for the Image Editing model.

### A.2 DATA FILTERING

To ensure dataset quality, we implement a multi-stage filtering pipeline to remove failed cases from the rendered images across different poses:

**Stage 1: Garment Detachment Check.** We examine the garment mesh from the simulation to detect garment slippage in the final frame. Specifically, we filter out cases where the highest point of the garment mesh is below the height of the SMPL root node, indicating garment detachment.

**Stage 2: Visual Plausibility Assessment.** We employ Qwen3-max API (Wang et al., 2024a) to assess the visual plausibility of rendered images. The VLM filters out implausible cases such as garment slippage, exposure of private body parts, or missing garment components. The text prompt used is shown in Fig. 8.

```
Evaluate whether this image shows a person wearing clothing
appropriately and naturally. Please check for the following
issues: (1) garment slippage or falling off the body, (2)
inappropriate exposure of private body parts, (3) missing or
incomplete garment components. Based on your assessment, answer
'acceptable' if the clothing appears normal and appropriate, or
'unacceptable' if any of these issues are present.
```

Figure 8: VLM evaluation prompt for quality filtering.

**Stage 3: Garment Coverage Verification.** We use Grounded-SAM (Ren et al., 2024) with "clothing" as the text prompt to verify that the segmented garment region adequately covers the human body. We filter out cases where the Intersection over Union (IoU) between the garment mask and the human body mask is below 50%.

**Validation.** To validate that our RealGarment dataset closely approximates real photo inputs, we conduct two validation studies. First, we randomly sample 1% of the generated rendered images for manual inspection, confirming a 99% rendering success rate. Second, we design a user study to evaluate garment-pattern alignment quality between photorealistic human images and garment pattern renderings. We randomly sample 1,000 images and ask human evaluators to assess alignment acceptability both before and after applying our filtering pipeline. Results show that our filtering pipeline significantly improves alignment acceptability from 64.3% (743/1000) to 99.6% (996/1000), demonstrating a substantial improvement of 35.3 percentage points in data quality.

### A.3 DATASET STATISTICS

Based on our proposed data curation pipeline, we synthesize RealGarment-1M (RG-1M), a dataset of over one million photorealistic human image-GarmentCode pairs. Combined with RGT-164K, our datasets provide diverse textures, poses, and photorealistic appearances for training robust image-conditioned GarmentCode generation models. Sample examples from the RealGarment-1M dataset are shown in Fig. 5. Detailed statistics of texture-related datasets are shown in Tab. 4.

## B MORE RESULTS

We present results generated by GarmentGPT. Using real human photographs as input, the corresponding original SMPL renderings are shown in the first row of each case. Please refer to Fig. 9, Fig. 10, and Fig. 11. Fig. 11 demonstrates GarmentGPT's results on inputs with complex textures.

## C DETAILED QUANTIZATION OF SEWING PATTERNS

Through a comprehensive analysis of sewing patterns, we discovered that their structure consists of multiple panels interconnected by stitching relationships among various line segments. Each panel

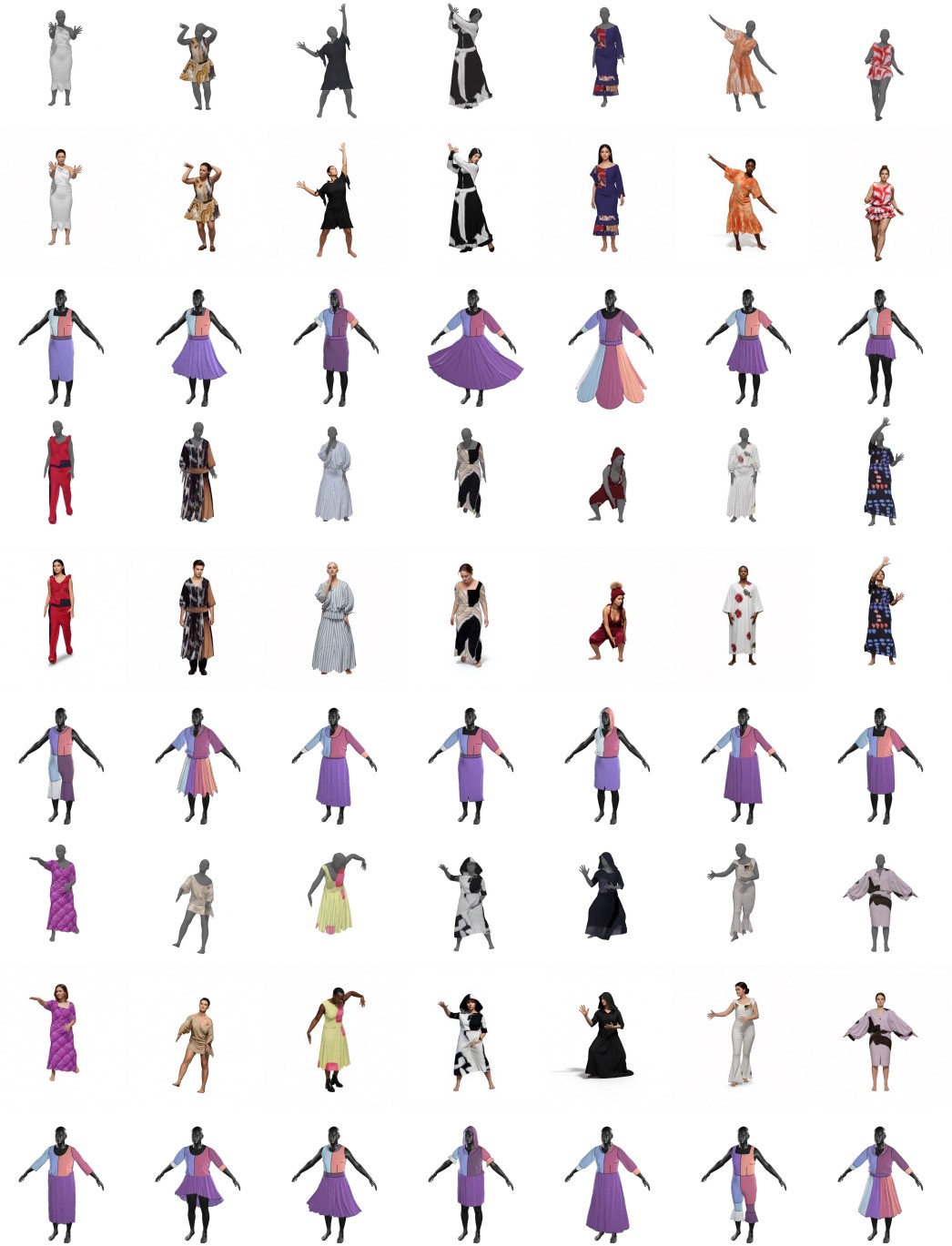

Figure 9: We show the results generated by GarmentGPT. Real human photos are used as input, with the corresponding original SMPL renderings shown in the first row of each case.

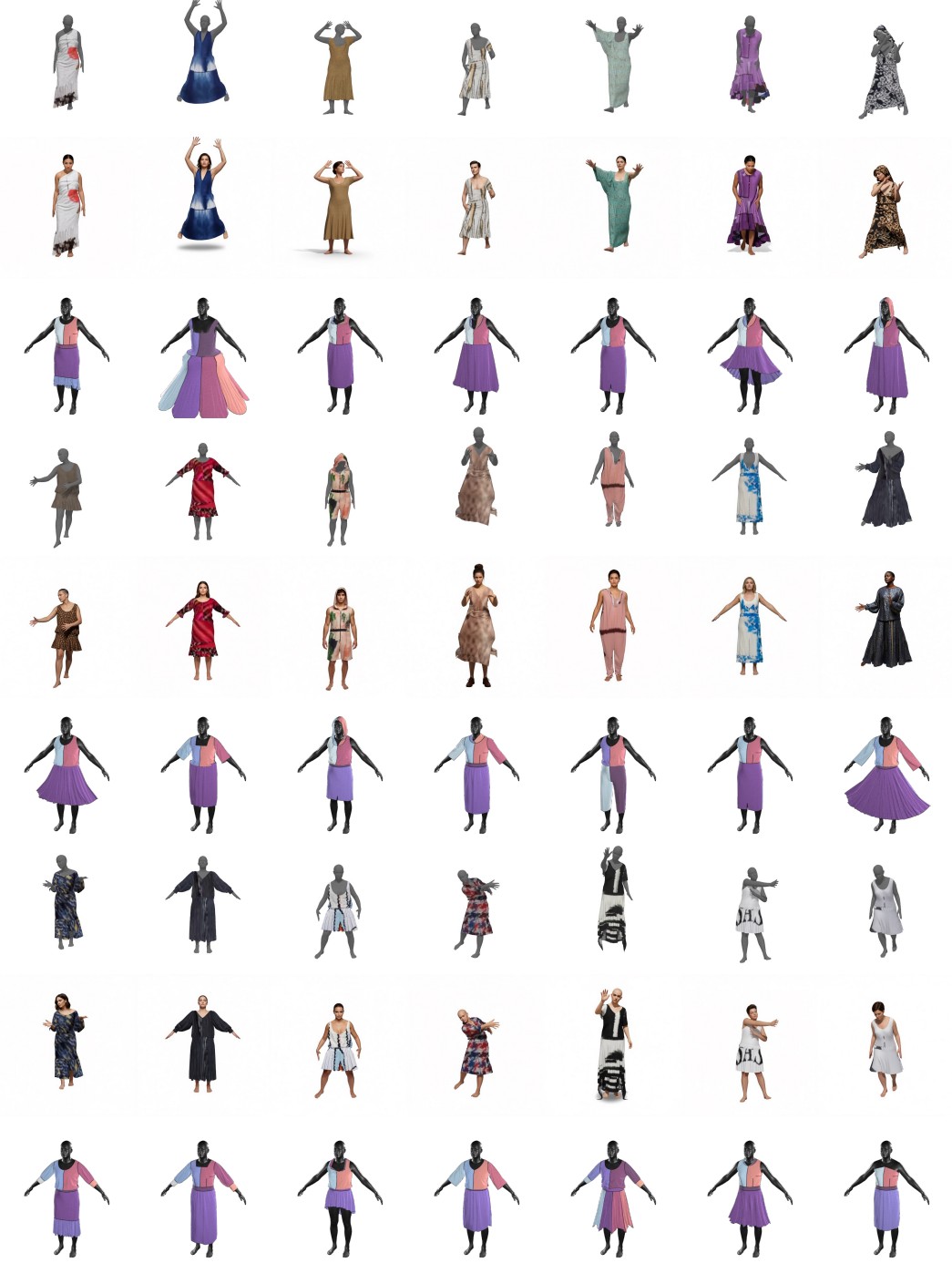

Figure 10: We show the results generated by GarmentGPT. Real human photos are used as input, with the corresponding original SMPL renderings shown in the first row of each case.

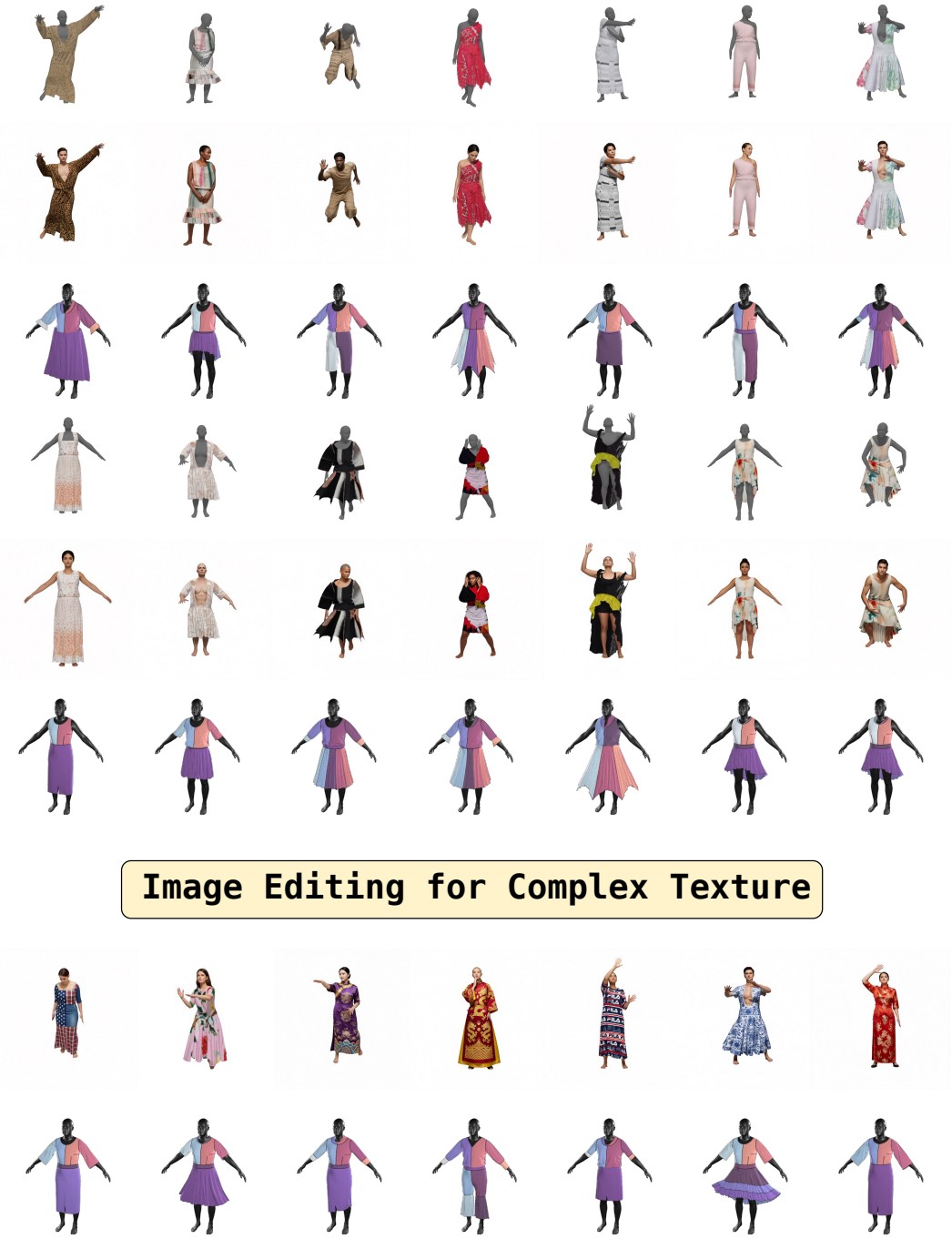

Figure 11: We show the results generated by GarmentGPT using real human photos as input. The corresponding original SMPL renderings are shown in the first row of each case. GarmentGPT maintains high success rates even with complex texture inputs.

Table 4: Comparison between clothing texture-related datasets and our proposed RealGarmentTexture-164K dataset

| Dataset | Type | Size | Description |
|---|---|---|---|
| **Texture Datasets** | | | |
| Textures Dataset (Huang et al., 2020) | Texture | 8.7K | General texture images |
| DTD (Cimpoi et al., 2014) | Texture | 5.6K | Describable textures database |
| TFD (Shakir & Topal, 2022) | Texture | 3K | Ten fabric categories |
| FID (Liu et al., 2024a) | Texture | 12.2K | Wool fabric |
| **RealGarmentTexture-164K (Ours)** | Texture | 164K | Garment Texture |
| **Garment Datasets** | | | |
| Clothes Dataset (chussboi96) (chussboi96, 2024) | Garment | 0.1K | Clothing images |
| Clothes Dataset (ryanbadai) (Badai, 2023) | Garment | 7.5K | Clothing images |
| Clothes Dataset (agrigorev) (Grigorev, 2020) | Garment | 7.5K | Clothing images |
| Dress Code (He et al., 2024) | Garment | 53.8K | Clothing images |
| **Fashion with Human Body Datasets** | | | |
| DeepFashion (Liu et al., 2016) | Human | 44K | Full-body fashion images |

possesses a unique identifier, spatial positioning in A-pose configuration (encompassing translation and rotation parameters), and a contour formed by consecutively connected closed edges. Each edge can be classified into four fundamental geometric types: straight lines, quadratic Bézier curves, cubic Bézier curves, and circular arcs. All edges share vertex parameters as foundational geometric information; quadratic and cubic Bézier curves incorporate one and two control point parameters, respectively; while circular arcs additionally encompass radius, drawing direction, and major/minor arc indicators as attributes. Through systematic encoding and high-precision reconstruction of this information, complete garment patterns can be faithfully reconstructed.

In our methodology, the quantization framework constitutes a comprehensive processing pipeline comprising three core modules: encoder, codebook, and decoder, designed to achieve efficient discrete representation and reconstruction of edge information and RT (translation-rotation) parameters within garment patterns.

To preserve the representational purity of distinct modal information, edges and RT parameters employ two independent sets of encoder-decoder pairs and codebooks, ensuring that the two information types remain mutually non-interfering during encoding and reconstruction processes. For edge information, to accommodate real-world design scenarios where precise control points are frequently absent, we uniformly sample each edge into N points as input. Regarding model architecture, we employ a ResNet-based edge encoder, whose lightweight, flexible, and efficient characteristics are particularly suitable for rapid training and generalization under large-scale datasets. For each panel's RT parameters, we concatenate translation and rotation information into a unified vector serving as RT encoder input. The RT encoder similarly adopts ResNet architecture, but with significantly fewer parameters compared to the edge encoder, matching its relatively lower data complexity.

Following the acquisition of encoded continuous representations, we map them to discrete indices through codebook mechanisms, thereby achieving effective quantized representations. We maintain independent codebooks for edges and RT parameters separately, and transform continuous latent vectors into discrete symbols through lookup table operations. Specifically, we employ Residual Quantization methodology to construct codebooks, utilizing multiple hierarchical indices to jointly represent an edge or RT parameter, where codebook vectors corresponding to each index exhibit residual relationships. By adjusting the number of residual hierarchy levels, we can control the representational capacity of latent vectors at different granularities, thus achieving a flexible balance between compression efficiency and reconstruction quality.

Ultimately, during the decoding stage, the system retrieves quantized latent vectors by querying discrete indices and reconstructs original parameters through symmetric ResNet decoders corresponding to the encoders. For edges, the decoder restores endpoint parameters and processes remaining attributes according to curve type distinctions: circular arcs require prediction of radius, drawing direc-

tion, and major/minor arc characteristics, while Bézier curves simplify parameter prediction through lossless conversion from quadratic to cubic forms. RT parameters directly regress to their original continuous values, thereby accomplishing high-fidelity reconstruction from discrete representation to complete pattern structure.

## D  THE USE OF LARGE LANGUAGE MODELS (LLMS)

Large language models (LLMs) were used only to assist with language editing and minor text improvements in the preparation of this manuscript. They were not involved in the design of the research, the development of methods, the execution of experiments, or the interpretation of results. All scientific content, including analyses, conclusions, and contributions, remains the work of the authors.

