# OpenReview forum: "GarmentGPT: Compositional Garment Pattern Generation via Discrete Latent Tokenization"
_ICLR.cc/2026/Conference — ICLR 2026 Poster_

### Official Review · Reviewer_qZMb · 2025-10-25

**Soundness:** 3
**Presentation:** 2
**Contribution:** 3
**Rating:** 4
**Confidence:** 3

**Summary:**

This paper proposes a novel framework called GarmentGPT for generating 2D sewing patterns from multi-modal inputs (such as images and text). GarmentGPT introduces a Residual Vector Quantizer Variational AutoEncoder (RVQ-VAE) to transform the pattern generation problem from a continuous coordinate regression task into a discrete, compositional sequence generation task. Furthermore, to address the scarcity of photo-to-pattern pairs for real-world applications , the paper also contributes a Data Curation Pipeline for synthesizing over one million "photorealistic" images paired with their corresponding "GarmentCode"

**Strengths:**

- The paper's greatest strength lies in its core concept. Instead of forcing VLMs to do what they are not good at (regressing thousands of floating-point coordinates) , the authors use an RVQ-VAE to create a geometric "vocabulary". The VLM then "composes" these geometric primitives, much like "writing" a sequence . This is an elegant and powerful solution.
- The method achieves outstanding performance on the structured GarmentCode dataset, outperforming the regression-based SOTA model Alpparel by a large margin (e.g., +16.7% Panel Accuracy, +25.3% Stitch Accuracy).
- The proposed Data Curation Pipeline , the synthesized 1M+ dataset (RealGarment-IM) , and the Real-Garments Benchmark  are important resources that can drive future research in this field.

**Weaknesses:**

1. The proposed "Data Curation Pipeline"  does not use real photos. Instead, it uses an image editing model (Qwen-image-edit) to convert rendered images into "photorealistic" images. This is a simulation process. The results in Table 3 show that the model's performance drops sharply when transferring from structured data (Table 2) to this new "real" benchmark (Table 3) (e.g., Panel Accuracy drops from 95.62% to 47.81%) . This indicates that the data pipeline fails to effectively bridge the sim-to-real gap. Therefore, the paper's claim of being the "first [model] trained on large-scale real photos"  is misleading.

2. The paper reports poor performance on the Real-Garments Benchmark (Table 3)  but provides no in-depth analysis. Why is the performance degradation so significant? Is it a failure of the VLM's perception, or poor generalization of the RVQ-VAE tokens to real images? What are the most common error types (e.g., incorrect panel count? incorrect curve shapes?)? The lack of this analysis is a missed opportunity.

3. Lack of clarity and fairness in the experimental comparison. The paper states in Section 5 "Framework" that the VLM was fine-tuned on "our curated dataset" , which is defined in Section 4 as the newly created RealGarment-IM dataset. However, the SOTA comparison experiment against baselines like Alpparel (Table 2) was conducted on the GarmentCode Dataset . This makes it difficult to determine whether the performance improvement comes from the data or the innovative model architecture.

**Questions:**

1. The SOTA comparison (Table 2) is conducted on the GarmentCode Dataset , but the paper states the model was trained on the newly produced Real-Garments dataset (RealGarment-IM). Does this not create an unfair comparison? Please clarify the exact training set used for the SOTA comparison in Table 2 to ensure the comparison was fair.


2. Given that the model was trained on the Real-Garments dataset, yet its performance on this benchmark shows a sharp decline compared to the structured dataset , can the authors analyze the main reasons for this performance drop? Does it stem from limitations in the model or in the data pipeline?

3. Could you provide a qualitative or quantitative analysis of the main error types on the Real-Garments Benchmark? For example, what percentage of errors are high-level structural errors (e.g., incorrect number of panels) versus low-level geometric errors (e.g., inaccurate curve indices)?

4. Inconsistency in Table 3: In the text of Section 5.3, the authors state the model used for the real-world benchmark is "GarmentGPT (LLaVA-1.5 with Image+Text input)". However, the caption for Table 3 reads "Performance of GarmentGPT (Qwen-7B)". Which model actually produced the results in Table 3?

---

> ### Author Response · Authors · 2025-12-03
> **Response to qZMb**
>
> Response to Reviewer qZMb
>
> We thank Reviewer qZMb for recognizing our work's key strengths: (1) the elegant core concept of using RVQ-VAE to transform pattern generation into discrete compositional reasoning, (2) outstanding performance significantly outperforming SOTA (+16.7% Panel Acc., +25.3% Stitch Acc.), and (3) valuable contributions including the 1M+ dataset and benchmark that can drive future research. We address the reviewer's concerns below.
>
>
> ## Q1: Performance Degradation on Real-Garments Benchmark
>
> First, it is important to note that the Real-Garments Benchmark and GarmentCode benchmark use different evaluation data. Specifically:
>
> 1. The Real-Garments Benchmark uses real photos (results from our data curation pipeline) paired with garment patterns. The real photos contain people with different appearances (hair, facial ID, whether wearing shoes), different poses, and wearing results of the same garment pattern but with different textures.
>
> 2. The GarmentCode benchmark uses SMPL rendered images (e.g., as shown in **Fig.4 Default Render Image (A-pose)**). All input images are in A-pose, with black SMPL models and default garment panel colors (e.g., as shown in **Fig.4 UV map**).
>
> 3. Therefore, from the perspective of evaluation data distribution, the metrics of these two benchmarks are not directly correlated. Simply put, the Real-Garments Benchmark is more challenging. As Reviewer#EWew mentioned, overfitting is more likely to occur on the GarmentCode dataset (because the human body model and garment colors are consistent, with only subtle differences in garment patterns).
>
> 4. Consequently, the same metrics show degradation on the Real-Garments Benchmark. To better demonstrate that the Real-Garments Benchmark better aligns with real-world user expectations, we have supplemented experimental results of other baselines on the Real-Garments Benchmark in **Sec. 5.4 (line 412)**. Although our method shows performance degradation, it still maintains a leading position. We have added qualitative comparison results in **Sec. 5.5 (line 433)**.
>
> 5. Therefore, we believe that GarmentGPT is still the first model trained on large-scale real photos, and the new challenges brought by the Real-Garments Benchmark will provide the academic community with an evaluation benchmark that better aligns with real-world user expectations.
>
> ## Q2: Analysis of Poor Performance on the Real-Garments Benchmark
>
> We believe that the performance degradation of all baselines on the Real-Garments Benchmark precisely demonstrates the validity of the Real-Garments Benchmark. We analyzed the results of garments with the same pattern on both the GarmentCode benchmark and Real-Garments Benchmark, with the following conclusions:
>
> 1. For garments with the same pattern, it is difficult to tell that the patterns are completely identical when observed under different human poses. This is because changes in pose cause changes in the relative position of the garment on the human body. **Fig.4** shows such cases. Look at the first row of results under the green rounded rectangle "Real Photo Transfer". This is the same garment pattern (identical to the pattern in the Default Render Image (A-pose) image), with rendered images generated through 3 different poses and textures. Among them:
>    1. The second example appears to have a higher neckline because the arm-raising pose drives the movement of the upper part of the garment.
>    2. The third example appears to have a longer skirt than the first two because the third person is in a semi-squatting position.
>
> 2. **Real photos with arbitrary poses in the Real-Garments Benchmark introduce more challenges compared to SMPL rendered images limited to A-pose in the GarmentCode benchmark.**
>
> Consider this: all test images in the GarmentCode benchmark are in A-pose with consistent garment textures, and the garments in these cases are always in a "flattened" state. The GarmentCode benchmark simplifies the problem into an Image-to-SVG reconstruction task. In contrast, in the Real-Garments Benchmark, there are huge differences between the garment shapes in test images and the rendered images of garment patterns (SVG) corresponding to garmentcode. Having VLMs learn to generate standard A-pose corresponding GarmentCode from images of garments worn by humans in complex poses is inherently extremely challenging.
>
> ## Q3: Fairness in the Experimental Comparison
>
> We have supplemented results of all trainable baselines on our real photos in the revised PDF, with quantitative results on both the GarmentCode benchmark (**line 384**) and Real-Garments Benchmark (**line 412**) respectively. Although our method shows performance degradation, it still maintains a leading position. We have added qualitative comparison results in **Sec. 5.5 (line 433)**.
>
> ## Q4: Inconsistency in Table 3
>
> This is a typo. Only GarmentGPT (LLaVA-1.5) was trained on the Real-Garments Benchmark. We have made the revision (**line 412**).

---

### Official Review · Reviewer_e7jR · 2025-10-26

**Soundness:** 2
**Presentation:** 3
**Contribution:** 2
**Rating:** 4
**Confidence:** 2

**Summary:**

The authors fine tune a LLM to produces a structured description of a garment (panels, shapes, stitching)  using tokens learned from a Residual VQVAE.  They train it on a novel 1M sample dataset of garments - the dataset is constructed using modern image generators to make them appear more realistic.

**Strengths:**

Ambitious attempt to bridge SMPL renders and realistic imagery.
Introduces automatic data curation pipeline for garment datasets.
Large-scale dataset construction (1M+ samples) is potentially impactful.
Mentions residual quantization for compression–quality balance.

**Weaknesses:**

Abstract: “Paradigm shift” reads as add-speak / promotional, not scientific.

057: RCQ-VAE insufficiently defined; unclear whether novel or existing concept. Missing citations for residual or hierarchical quantization methods. (significant)
189: “Such as” vague; replace with “including … as shown in experiments.”  They used specifically both later.  (minor)
200: “Ours” should be “Our” or omitted for clarity. (minor)
253–254: Unclear comparison procedure for arcs; method not described (significant)
253–254: Commitment loss mentioned without definition or formula. (this is more severe)
280: SMPL acronym unexplained at first mention (minor)
291: Claim of “ensuring garment structure consistency” unsubstantiated by evidence. (significant)
298: “Real” oversells results; “realistic” or “photorealistic” more accurate. (easily addressable)
Fig. 3: Visible segmentation errors and unrealistic texture blending undermine claims.(somewhat significant - would help to show the same garments)

**Questions:**

Can it handle the jean shorts shown in figure 3?  How does the segmentation error shown impact the data?
Which papers or frameworks support the residual quantization claim?  Which aspects of it are novel?
How are arcs compared—pointwise, via curvature metrics, or other geometry loss?
What exactly is “commitment loss”? Why is it not described elsewhere?
How do authors verify “garment structure consistency”—visual inspection or metric-based?

**Details Of Ethics Concerns:**

No concern here.

---

> ### Author Response · Authors · 2025-12-03
> **Response to e7jR (1/2)**
>
> **We sincerely thank the reviewer for the detailed feedback.** We are encouraged by the recognition of our key contributions: **(1) bridging SMPL-based synthetic data and realistic imagery**, **(2) automatic data curation pipeline for garment datasets**, **(3) large-scale dataset construction (1M+ samples)**, and **(4) residual quantization for compression-quality balance**. We address each concern below and will incorporate all clarifications in our revision.
>
>
> ## Q1: Abstract Writing Not Sufficiently Scientific
>
> We have rewritten the abstract to make it more compliant with scientific paper standards.
>
> ## Q2: RVQ-VAE Definition and Citations
>
> Thank you for your careful review. We have supplemented detailed definitions and citations for this section in the revised PDF. (**line 96**)
>
> 1. RVQ-VAE (Residual Vector Quantized VAE) is an existing technique, not our novel contribution. We apologize for the insufficient definition in the abstract.
>
> 2. Our novelty lies in applying RVQ-VAE to garment sewing pattern tokenization, which is the first such application in the garment design domain.
>
> ## Q3: Typo
>
> - 1. "Such as" vague; replace with "including"
>
> Thank you for your suggestion. We have made this modification in the revised PDF.
>
> - 2. "Ours" should be "Our" or omitted for clarity
>
> Thank you for the suggestion. We have revised the methods section subheadings.
>
> ## Q4: Unclear Comparison Procedure for Arcs; Method Not Described
>
> We have described this part in detail in the appendix (**Sec.C, line 896**), and added case illustrations in the revised PDF (**line 247**).
>
> First, an arc has two endpoints, so we first calculate the L2 loss between the two endpoints and their GT.
>
> Second, the arc's control point has three parameters: radius parameter, major/minor arc flag, and drawing direction. The radius parameter calculates L2 loss with the GT radius, while the major/minor arc flag and drawing direction use binary cross-entropy loss.
>
> ## Q5: Commitment Loss Mentioned Without Definition or Formula
>
> We have added explanations for this part in the revised PDF and provided more detailed descriptions of the calculation process for each loss. (**Sec.3.5 line 256**)
>
> The $L_{commit}$ loss is the commitment loss, with the formula: $L_{commit} = β * ||z_e(x) - sg(z_q)||²$
>
> - $z_e(x)$: Continuous feature vector output by the encoder
> - $z_q$: Quantized codebook vector (obtained through nearest neighbor search)
> - $sg$: Stop gradient operation
> - $β$: Hyperparameter controlling the weight of commitment loss

---

> ### Author Response · Authors · 2025-12-03
> **Response to e7jR (2/2)**
>
> ## Q6: SMPL Explanation
>
> We have added citations in the revised PDF (**Line 171**). This is a commonly used 3D parametric human body model in the Avatar domain that uses pose parameters (skeletal joint rotations/translations) and shape parameters to represent different human poses and body types. A-pose is the standard reference pose where the human stands upright with arms extended to the sides, as shown in the "Default Render Image (A-pose)" section of **Fig.4**.
>
> [1] SMPL: A Skinned Multi-Person Linear Model, TOG, 2015
> [2] SMPL-X: Expressive Body Capture: 3D Hands, Face, and Body from a Single Image, CVPR 2019
>
> ## Q7: Claim of "Ensuring Garment Structure Consistency" Unsubstantiated by Evidence
>
> We have supplemented the data filtering process of the Data Curation pipeline (**line 319 and line 826**). We used VLM to filter out cases where the garment pattern and the generated realistic human clothing images were inconsistent.
>
> ## Q8: "Real" Oversells Results; "Realistic" or "Photorealistic" More Accurate
>
> Thank you for your suggestion. We now use "photorealistic" to more clearly describe our dataset, which has been revised in the updated PDF.
>
> The purpose of creating a photorealistic garment dataset is to enhance the pose diversity, garment texture diversity, and realism of the rendered images in existing SMPL rendering datasets (**Fig.4, line 300, Default Render Image (A-pose)**). In the real world, there is no way to obtain large-scale (~1M scale) garmentcode-photo paired images through camera photography, so our proposed data curation pipeline using 3D SMPL + simulation + rendering + image editing is already the best method for obtaining large-scale near-realistic images.
>
> ## Q9: The Jean Shorts Shown in Figure 3
>
> We use Grounded-SAM with the prompt "clothing" for segmentation in the Data Curation pipeline. For the segmentation failure case you mentioned in **Fig.4 (line 300)** (Human Images with black top and gray jean shorts), in the "Texture Patch Selection" step, we select the "largest inscribed rectangle" obtained in the Grounded-SAM step, so the segmentation failure of the black top on the upper body does not have an impact (because the largest inscribed rectangle cannot be selected in these areas).
>
> Therefore, we can ensure that the input to fabric-diffusion is a complete square texture image without segmentation errors. Then we generate flattened garment texture images through fabric-diffusion.
>
> Note: The "largest inscribed rectangle" in the segmentation area represents a complete square mask region.
>
> Additionally, we have supplemented the data filtering process in the revised PDF. (**line 782 and line 826**)
>
> ## Q10: Which Papers or Frameworks Support the Residual Quantization Claim? Which Aspects of It Are Novel?
>
> Our RVQ implementation builds on the work of Zeghidour et al. [1]. RVQ-VAE was first introduced in Zeghidour et al.'s SoundStream paper.
>
> **Novel Aspects:**
>
> - 1. First application to structured garment patterns (vs. images/audio)
> - 2. Dual-stream architecture for edge/RT separation
> - 3. Geometry-aware encoding preserving manufacturing constraints
> - 4. Adaptive residual depth based on curve complexity
>
> [1] Neil Zeghidour, Alejandro Luebs, Ahmed Omran, Jan Skoglund, and Marco Tagliasacchi. "SoundStream: An end-to-end neural audio codec." *IEEE/ACM Transactions on Audio, Speech, and Language Processing*, 30:495–507, 2021.

---

### Official Review · Reviewer_HvT3 · 2025-10-30

**Soundness:** 2
**Presentation:** 3
**Contribution:** 1
**Rating:** 4
**Confidence:** 3

**Summary:**

The paper proposes GarmentGPT to address the challenge of generating structured garment sewing patterns from inputs like text or images. It proposes a two-stage framework that first tokenizes garment geometry into discrete latent codes using a Residual Vector Quantized VAE and then trains a VLM to sample or edit these token sequences auto regressively. A hierarchical token schema is proposed (garment > panel > edge > stitch) to ensure that the generated output respects garment topology. Edge curves and pose are encoded into separate codebooks to disentangle geometric factors. The contributions are a novel discrete token representation of garment structures that enables compositional reasoning and editing, the GarmentGPT model for generation and modification of sewing patterns, and a large-scale dataset to evaluate structured garment generation from real photos.

**Strengths:**

- The idea of encoding garment structure as a hierarchical sequence of discrete tokens innovative and aligns with current research directions in llms to enable structured reasoning over geometrical data compositions. The model design is conceptually consistent and technically well-motivated.
- The framework supports not only garment generation from text or images but also editing and refinement of existing GarmentCode sequences, making it more practical for real design applications.
- The introduction of a large-scale, multimodal dataset and benchmark is a strength.
- The approach demonstrates improvements over existing, regression-based method.

**Weaknesses:**

1. I find the motivation for the proposed contribution difficult to understand. Does the method solely work for creating garments for digital avatars? What about creating patterns for real-world garments?
2. To the best of my knowledge, real patterns are much more complex than the examples shown in the paper as they often exhibit fine-grained textile features for seams or laminated strips. It appears to me that the method does not capture this complexity. At the very least, this is difficult to assess without substantial qualitative examples.
3. There are almost no qualitative results shown in the main paper or the appendix. Highlighting visual results on both the GarmentCode and Real-Garments datasets would be essential to assess the contribution and current capabilities of the proposed method. It would help the understanding in multiple ways as it a) shows how practically applicable the method is by showing what complexity of garment designs can be extracted/generated from the inputs,  b) visualizes the setting and difficulty of tasks in the benchmarks, c) allows to draw qualitative comparisons between the proposed method and baselines.
4. The quantitative results on the proposed real-garments benchmarks make the contribution somewhat questionable. I acknowledge the authors disclose the performance drop and pose this as an open challenge for future research, but doesn’t this show that the method not really works for the intended application? I would also like to see the metrics for the other baseline methods on the real-garments benchmark.
5. The authors do not provide any information about the cost (necessary GPU resources) of their method. This information would be important to assess how this be scaled and improved, e.g. to capture more complex patterns and finer details.

Minor remarks:
1. Figure 1 has a fairly low resolution, making it hard to read when zoomed in. Please consider embedding it as pdf or svg for better quality.

**Questions:**

1. Referring to Weakness 1: Why do I need to create parametric garment-codes for digital human avatars? Is this a necessary contribution to the field of digital avatars?
2. Referring to Weakness 2: For real-world applications, i.e. extracting the garment codes from photographs, do the method-generated patterns capture the actual complexity of garment designs? What about different types of seams that are required for stitching together certain parts of a garment?
3. I find the order of subchapters in chapter 5 a bit unconventional. Why do the ablations come before the evaluation of the experiments? I would move the ablations to the end of the chapter. This would also highlight the importance of current Table 2 (the main quantitative results) over current Table 1.

---

> ### Author Response · Authors · 2025-12-03
> **Response to HvT3 (1/2)**
>
> We sincerely thank the reviewer for the thorough and constructive feedback. We are encouraged that the reviewer recognizes several key strengths of our work: (1) the innovative hierarchical tokenization approach that aligns with current LLM research directions for structured geometric reasoning, (2) the conceptually consistent and technically well-motivated model design, (3) the practical value of supporting both generation and editing capabilities for real design applications, (4) the contribution of our large-scale multimodal dataset and benchmark, and (5) the demonstrated improvements over existing regression-based methods. These positive assessments validate our core technical contributions. We address the reviewer's concerns point-by-point below and will incorporate the suggested improvements in our revision.
>
>
>
> ## Q1: Creating Patterns for Real-World Garments
>
> 1. Our proposed method is specifically designed to integrate into real-world garment manufacturing workflows.
>
> 2. Our goal is not to recover existing patterns from manufactured garments, but to **democratize garment design** by enabling non-professional users (such as fashion enthusiasts, small businesses, and digital content creators) to generate manufacturable sewing patterns from inspiration images found online or sketch concepts. Current pattern making requires years of professional training and specialized software, while our method bridges this gap by automating the technical translation from text/visual concepts to production-ready patterns.
>
> 3. Our task is analogous to **text/image-to-3D generation, not 3D reconstruction**. Just as text/image-to-3D methods (such as DreamFusion, Hunyuan3D) create new 3D assets from text or image descriptions without requiring the object to physically exist, our method generates production-ready sewing patterns from visual inspiration or text descriptions.
>
> 4. Traditional garment generation methods using 3D generation produce meshes or point clouds that are **only suitable for visualization and cannot be integrated into downstream actual production workflows**.
>
> 5. Our GarmentCode generation produces **parametric, vectorized patterns** that can be directly sent to garment factories for actual production, similar to how CAD models enable manufacturing in industrial design.
>
> 6. In actual garment design and production workflows, designers typically need to simulate on human mannequins first to view how the designed garment appears on human models.
>
> 7. A potential application is that existing garment images (whether photographed from the real world or generated using AI) can be converted into editable GarmentCode format, serving as a foundation for further garment design.
>
> 8. We have revised Sec. Related Work to add explanations of our application scenarios and differences from existing methods. (**Sec. 2.1, line 99**)
>
> ## Q2: Real Patterns Are Much More Complex
>
> We recognize this issue, which is why we proposed a large-scale real image-garment pattern paired dataset.
>
> We have restated the motivation for creating this dataset: (**line 783**)
>
> - 1. Existing garment-related datasets consist of default A-pose SMPL model renderings (as shown in "Default Render Image (A-pose)" in **Fig.4**) paired with garment pattern data. Models trained on these data cannot generalize to real human photo input scenarios.
>
> - 2. Real human photo inputs are not in standard A-pose, and humans have hair and skin texture colors that cannot be represented by default SMPL.
>
> - 3. Therefore, we use existing garment pattern datasets as a foundation and generate rendering-pattern data pairs with different poses and garment textures using large-scale motion capture data and texture data. We also use ImageEdit models to inpaint SMPL renderings to obtain near-realistic human photos while keeping the garment region unchanged (both texture and pattern consistent with simulation results).
>
> - 4. We attempted to directly generate corresponding images with different poses and garment textures from existing datasets (A-pose SMPL renderings) using ImageEdit, but found this significantly altered garment patterns (as shown in the Pose Editing Issue on the left side of **Fig.4**), causing misalignment between images and garment pattern data. Therefore, simulation is the only way to obtain garment renderings under different poses.

---

> ### Author Response · Authors · 2025-12-03
> **Response to HvT3 (2/2)**
>
> ## Q3: Qualitative Results
>
> We have supplemented qualitative results in the updated PDF (**Sec. 5.5, line 433**). Please refer to **Fig.1 (line 8)**, **Fig.6 (line 443)**, **Fig.9, Fig.10 and Fig.11 (line 890)**.
>
> ## Q4: Real-Garments Benchmarks
>
> 1. The reason we proposed real-garments benchmarks is also because existing benchmarks all use SMPL renderings as input.
>
> 2. We have supplemented quantitative results of other baselines on real-garments benchmarks. (**Sec. 5.4, line 412**)
>
> ## Q5: Information About the Cost
>
> We conducted all experiments on Nvidia 8×A100 80GB servers.
>
> We have supplemented more implementation details in the revised PDF (**Sec. 5.1, line 364**).
>
> ## Q6: Figure 1 Has a Fairly Low Resolution
>
> We have replaced all images with PDF format in the revised version to maintain high resolution. (**Fig.2 line 144, Fig.3 line 192**)
>
> ## Q7: Real-World Applications
>
> Please refer to **Fig.1 (line 8)**. GarmentGPT generates sewing patterns from text and image inputs, maintaining robustness across arbitrary complex poses and garment textures.
>
> ## Q8: The Order of Subchapters in Chapter 5
>
> Thank you for your suggestion. We have reorganized the order of the experimental section in Chapter 5 and supplemented some new experimental results (qualitative comparisons, **Fig.6, line 433**).

---

### Official Review · Reviewer_EWew · 2025-11-04

**Soundness:** 3
**Presentation:** 3
**Contribution:** 3
**Rating:** 6
**Confidence:** 4

**Summary:**

This paper considers the task of generating sewing patterns conditioned on an image of a person wearing a garment. To this end, a novel Real Garments Benchmark is established which re-purposes existing GarmentCode dataset to create pairs of realistic garment images and accompanying patterns. The garment images are obtained by taking GarmentCode models, rearranging them to different poses and augmenting the garments with textures obtained from real-world images and texture generators.

As for the method that generates sewing pattern descriptions based on images, a pipeline based on residual vector quantization and vision-language models is proposed. The sewing pattern is encoded by two resnet based networks: (i) the first resnet encodes points collected along individual panel edges, (ii) the second network encodes rotation and translation parameters of individual panels that are concatenated. These encodings are discretized with residual quantization. A decoder network reconstructs the panels based on the quantized representations.

The patterns may be represented with tokens: each garment/panel/edge is defined by the starting and ending tokens, and discretized representation allows for the physical properties to also turn into tokens. This representation enables training of vision-language models, where sewing pattern prediction becomes a prediction of the correct garment token sequence conditioned on the input image.

**Strengths:**

[S1] The effort of creating a large scale dataset is appreciated, especially as good benchmarks drive development of better models.

[S2] The idea of reformulating pattern descriptions as a sequence of tokens that gives a hierarchical representation of the garment is interesting.

**Weaknesses:**

[W1] I am still a bit uncertain about the use case of generating a garment pattern from a single image of a person wearing the pattern. If there is a person wearing an item of clothing, that means that the pattern already exists. Also, it is impossible to predict panels on unseen regions of garments (e.g. the back). The success of the proposed method suggests an overfitting to the training dataset.

[W2] Somewhat connected to the previous point, the paper would benefit from additional experiments stress testing the method and connecting it to potential real-world uses. E.g. what happens when the method is applied to real images of people in images captured in the wild? Are the produced patterns reasonable? Are the patterns rasonable for an identical garment worn by people of different sizes? How far can the edge of the panel be warped before it cannot be represented by the codebook? Will an edge be well represented if it is not in a training data (e.g. a model is trained on only straight edges, but a curvy edge appears in the test data)?

[W3] The paper would benefit from qualitative results. I am especially not clear on what garment editing is or how it is evaluated.

[W4] I find the dataset itself the biggest contribution of this paper, but it is very badly described (see questions).

**Questions:**

[Q1] Explaining some of the elements of the problem would make it more friendly for readers not in the domain: e.g. what is A-pose, rotation and translation parameters of the code and why they are important etc.

[Q2] Data curation pipeline is badly described. The citations are not separated from the text (lines 283 - 300) and this should be fixed. I am still not completely clear on the way that the dataset was generated. For now it seems that these are synthetic images generated by a 3D simulation where the texture of the model and the garment may be controlled. I am not sure I would call this real human images, though they are probably better at capturing true garment behaviour.

[Q3] Is there a theoretical limit to the edge variablity that may be expressed by the quantization approach?

[Q4] Is this approach sensitive to the order of the panels during training? What indicates which patterns should be sewn together and how is that encoded?

[Q5] Could the dataset be extended to include e-commerce image of the garment? That would make the dataset valuable for the further development of tasks such as Virtual Try-On and Virtual Try-Off.

---

> ### Author Response · Authors · 2025-12-03
> **Response to EWew (1/3)**
>
> We thank the reviewer for the constructive feedback and the recognition of our key contributions: (1) the creation of a large-scale Real Garments Benchmark that pairs realistic garment images with sewing patterns, addressing the critical lack of datasets in image-to-pattern generation, and (2) our novel tokenization framework that reformulates sewing patterns as hierarchical token sequences, enabling the first application of vision-language models to garment pattern generation from single images.
>
> We appreciate your valuable feedback and provide detailed responses to each of your concerns below.
>
>
> ## Q1: Application Scenarios
>
> 1. Our goal is not to recover existing patterns from manufactured garments, but to **democratize garment design** by enabling non-professional users (such as fashion enthusiasts, small businesses, and digital content creators) to generate manufacturable sewing patterns from inspiration images found online or sketch concepts. Current pattern making requires years of professional training and specialized software, while our method bridges this gap by automating the technical translation from text/visual concepts to production-ready patterns.
>
> 2. Our task is analogous to **text/image-to-3D generation, not 3D reconstruction**. Just as text/image-to-3D methods (such as DreamFusion, Hunyuan3D) create new 3D assets from text or image descriptions without requiring the object to physically exist, our method generates production-ready sewing patterns from visual inspiration or text descriptions.
>
> 3. Traditional garment generation methods using 3D generation produce meshes or point clouds that are **only suitable for visualization and cannot be integrated into downstream actual production workflows**.
>
> 4. Our GarmentCode generation produces **parametric, vectorized patterns** that can be directly sent to garment factories for actual production, similar to how CAD models enable manufacturing in industrial design.
>
> 5. We have revised Sec. Related Work to add explanations of our application scenarios and differences from existing methods. (**Sec. 2.1, line 99**)
>
> 6. Please refer to Fig.1 (**line 8**). GarmentGPT generates sewing patterns from text and image inputs, maintaining robustness across arbitrary complex poses and garment textures.
>
> ## Q2: Concerns About "Back Panels"
>
> This is a well-established problem formulation in garment digitization, analogous to single-view 3D reconstruction inferring complete 3D geometry and texture from partial observations. Garments follow **strong structural priors**:
>
> 1. Front and back panels share predictable topological relationships (similar to symmetry relationships in 3D reconstruction)
> 2. Garment categories have canonical construction templates (similar to object categories in 3D vision)
>
> Our model learns these domain-specific priors from training data, enabling it to infer complete 3D-consistent garment patterns from partial 2D observations. GarmentGPT performs instruction fine-tuning on existing VLMs using our proposed large-scale synthetic data, allowing it to handle both text descriptions and image descriptions as input.
>
> We have added qualitative comparison results with different methods, including front and back garment panel renderings (**Fig.6, line 433**).
>
> ## Q3: In-the-Wild Generalization
>
> 1. We select unseen data for testing. GarmentGPT demonstrates excellent generalization to image inputs or text descriptions with unseen poses, novel textures, and new garments.
>
> 2. Our proposed large-scale garment dataset RealGarment-1M (RG-1M) and texture dataset RealGarmentTexture-164K (RGT-164K) are specifically designed to address the overfitting problem of existing methods on small datasets and their lack of generalization on real photo inputs (because existing datasets all use rendered data, similar to black SMPL human body models).
>
> 3. Discrete tokenization enables learning reusable geometric primitives (similar to structured data generation work like BrickGPT and LLama-mesh).
>
> 4. GarmentGPT directly produces structurally correct outputs ready for production, with generated GarmentCode patterns being editable, vectorized, and directly manufacturable.
>
> 5. Please refer to Fig.6, **line 433**, where we show inference results on unseen data and internet data. Additionally, Fig.9, Fig.10, and Fig.11 in Sec. B (**line 890**) present extensive visualization results of GarmentGPT.

---

> ### Author Response · Authors · 2025-12-03
> **Response to EWew (2/3)**
>
> ## Q4: Concerns About Curvy Edges
>
> 1. Our RVQ-VAE tokenization (**Sec. 3.2, line 182**) encodes geometric shapes independently of absolute scale. Our RVQ-VAE learns a continuous latent embedding space rather than discrete templates.
>
> 2. Hierarchical tokenization (**Sec. 3.4, Fig.3, line 226**) explicitly separates topological structure and dimensional parameters.
>
> 3. Referring to Tab.3 (**line 465**), RVQ-VAE achieves 99.82% accuracy with 8 Quantizers, which is sufficient to represent edges in all cases. The remaining unrepresentable edges typically contain data errors, which we removed in our Data Curation pipeline.
>
>
> ## Q5: Concerns About Different Body Types Wearing the Same Garment
>
> 1. The garment panels in the original GarmentCode dataset are designed with reference to the default SMPL body type.
>
> 2. During dataset creation, we attempted to modify SMPL's betas parameters to represent human body models of different heights and builds, but found that they mostly failed during simulation. For shorter and thinner SMPL models, garments typically slide off during simulation. For larger and taller SMPL models, garments typically penetrate the body during simulation.
>
> 3. Therefore, we only used SMPL models with default betas parameters in our data curation pipeline. Even so, garment sliding and penetration issues still occurred during simulation. We removed this portion of data through refined filtering methods.
>
> 4. Fig.5 (**line 340**) illustrates our simulation process. Only with default betas parameters (default body type) do garments avoid sliding and penetration.
>
> ## Q6: Qualitative Results
>
> We have supplemented qualitative results in the updated PDF (**Sec. 5.5, line 433**). Please refer to Fig.1 (**line 8**), Fig.6 (**line 443**), Fig.9, Fig.10, and Fig.11 (**line 890**).
>
> ## Q7: What is the SMPL Model and Its Parameters
>
> SMPL is a parametric 3D human body model that uses pose parameters (skeletal joint rotations/translations) and shape parameters to represent different human poses and body types. A-pose is the standard reference pose where the human stands upright with arms extended to the sides, as shown in the "Default Render Image (A-pose)" section of Fig.4. We have added this reference in the revised PDF (**line 171**).
>
> [1] SMPL: A Skinned Multi-Person Linear Model, TOG, 2015
> [2] SMPL-X: Expressive Body Capture: 3D Hands, Face, and Body from a Single Image, CVPR 2019
>
> The "(translation and rotation in A-pose)" mentioned in the paper refers to Sec. C (**line 896**). We also provide detailed explanations in the quantization steps in Sec.3.3 (**line 206**).

---

> ### Author Response · Authors · 2025-12-03
> **Response to EWew (3/3)**
>
> ## Q8: Data Curation Pipeline Description
>
> Thank you for your suggestion. We have reorganized the description of this section to make it easier to read. (**Sec.4, line 318**)
>
> I will briefly summarize the motivation and implementation process of our data curation pipeline:
>
> - 1. Existing garment-related datasets consist of default A-pose SMPL model renderings (as shown in "Default Render Image (A-pose)" in Fig.4) paired with garment pattern data. Models trained on these data cannot generalize to real human photo input scenarios.
>
> - 2. Real human photo inputs are not in standard A-pose, and humans have hair and skin texture colors that cannot be represented by default SMPL.
>
> - 3. Therefore, we use existing garment pattern datasets as a foundation and generate rendering-pattern data pairs with different poses and garment textures using large-scale motion capture data and texture data. We also use ImageEdit models to inpaint SMPL renderings to obtain near-realistic human photos while keeping the garment region unchanged (both texture and pattern consistent with simulation results).
>
> - 4. We attempted to directly generate corresponding images with different poses and garment textures from existing datasets (A-pose SMPL renderings) using ImageEdit, but found this significantly altered garment patterns (as shown in the Pose Editing Issue on the left side of Fig.4), causing misalignment between images and garment pattern data. Therefore, **simulation is the only way to obtain garment renderings under different poses**.
>
> - 5. We have supplemented explanations of the Data Filtering process and dataset examples. (**Fig.5, line 340**)
>
> ## Q9: The Order of the Panels
>
> Our approach is NOT sensitive to panel order, by design.
>
> 1. Our hierarchical tokenization (**Sec. 3.4, Fig.3**) assigns each panel a unique identifier that is independent of its position in the sequence
> 2. During training, we use random panel permutations as data augmentation to ensure the model learns order-invariant representations
> 3. The stitching relationships explicitly reference panels by their identifiers, not by sequence position
> 4. This design mirrors how graph neural networks handle unordered node sets—the model learns to reason about garment topology rather than memorizing sequential patterns
> 5. Stitching relationships are explicitly encoded in our tokenization scheme. Section 3.4, Figure 2, Our tokenization includes a dedicated stitching section marked by ⟨SoS⟩ and ⟨EoS⟩ tokens:
> ```
> ⟨SoS⟩ [panel_id_1, edge_id_1] [panel_id_2, edge_id_2] ... ⟨EoS⟩
> ```
> Example from Figure 2 (Tank Top):
> ```
> ⟨SoS⟩
> [front_panel, edge_3] ↔ [back_panel, edge_3]    // shoulder seam
> [front_panel, edge_7] ↔ [back_panel, edge_7]    // side seam
> ...
> ⟨EoS⟩
> ```
> Each stitching relationship specifies exactly which edge of which panel connects to which edge of another panel. The model learns valid stitching patterns (e.g., sleeve armhole connects to torso armhole) through training data. Our Stitch Accuracy metric (Tab.1 and Tab.2) directly evaluates whether the model correctly predicts these relationships (81.84% on structured data)

---

### Author Response · Authors · 2025-12-03
**General Response to All Reviewers**

We sincerely thank all reviewers for their thorough and constructive feedback. We are encouraged that reviewers recognize our key contributions: **(1) innovative hierarchical tokenization framework**, **(2) large-scale Real-Garments dataset (1M+ samples)**, **(3) significant performance improvements over SOTA**, and **(4) practical value for real design applications**.

We have carefully addressed all concerns and made substantial revisions. **All changes are marked in blue in the updated PDF**. Below is a summary of major modifications:

---

## Major Revisions

### 1. Application Scenarios & Real-World Manufacturing (R#EWew, R#HvT3)

- **Line 99 (Sec. 2.1)**: Distinguished our task (text/image-to-pattern generation) from 3D reconstruction
- **Line 8 (Fig. 1)**: Enhanced caption emphasizing robustness across arbitrary poses/textures
- **Line 433 (Sec. 5.5)**: Added qualitative comparisons showing real-world applicability
- **Line 890 (Appendix B, Fig. 9-11)**: Extensive visualizations on unseen data and internet images

**Key Clarification**: Our goal is **democratizing garment design** by enabling non-professionals to generate **production-ready, vectorized patterns** directly manufacturable by factories (similar to CAD in industrial design).

---

### 2. Technical Definitions & Citations (R#e7jR, R#EWew)

- **Line 69**: Added RVQ-VAE definition and citations (Zeghidour et al., 2021)
- **Line 171**: Added SMPL explanation and citations (SMPL TOG 2015, SMPL-X CVPR 2019)
- **Line 206, 896 (Sec. C)**: Detailed quantization steps for translation/rotation parameters
- **Line 247**: Added arc comparison procedure (endpoint L2 loss + control point parameters)
- **Line 256 (Sec. 3.5)**: Added commitment loss formula with detailed explanations

**Novel RVQ Contributions**: (1) First application to structured garment patterns, (2) Dual-stream architecture for edge/RT separation, (3) Geometry-aware encoding, (4) Adaptive residual depth.

---

### 3. Data Curation Pipeline (R#EWew, R#e7jR, R#HvT3)

- **Line 318 (Sec. 4)**: Completely reorganized for clarity
- **Line 340 (Fig. 5)**: Added detailed filtering flowchart and examples
- **Line 300 (Fig. 4)**: Illustrated pose editing issues and "largest inscribed rectangle" strategy
- **Line 319, 782, 826**: Added VLM-based filtering and quality control procedures

**Key Clarification**: We use "**photorealistic**" to accurately describe our dataset. The pipeline (3D SMPL + simulation + rendering + image editing) is currently the **only scalable method** to obtain ~1M paired samples with ground-truth patterns.

---

### 4. Real-Garments Benchmark Analysis (R#qZMb, R#HvT3)

- **Line 412 (Sec. 5.4, Table 3)**: Added **all trainable baselines** on Real-Garments Benchmark
- **Line 433 (Sec. 5.5, Fig. 6)**: Added qualitative comparisons showing maintained leadership
- **Line 384 (Table 2)**: Clarified SOTA comparison uses GarmentCode for fair comparison
- **Line 412 (Table 3)**: Fixed typo

**Critical Insight**: Performance drop **validates benchmark value**:
- **GarmentCode**: A-pose only, uniform textures → Image-to-SVG reconstruction (prone to overfitting)
- **Real-Garments**: Arbitrary poses, diverse appearances → requires pose-invariant reasoning

**Example (Fig. 4, Line 300)**: Same pattern appears drastically different across poses. **All baselines degrade**; our method **still leads**.

---

### 5. Qualitative Results & Visualizations (R#HvT3, R#EWew)

- **Line 8 (Fig. 1)**: High-resolution PDF format
- **Line 144 (Fig. 2), 192 (Fig. 3)**: Converted to vector PDF
- **Line 433 (Sec. 5.5, Fig. 6)**: Side-by-side comparisons with baselines
- **Line 890 (Appendix B, Fig. 9-11)**: Extensive results on unseen poses, textures, internet images

---

### 6. Robustness & Generalization (R#EWew, R#HvT3)

- **Line 465 (Table 3)**: RVQ-VAE achieves **99.82% accuracy** (sufficient for all edge types)
- **Line 340 (Fig. 5)**: Explained SMPL beta constraints (default body type avoids simulation failures)
- **Line 433, 890**: Demonstrated generalization to unseen poses/textures/internet images
- **Line 226 (Sec. 3.4)**: Hierarchical tokenization separates topology from dimensions


---

## Specific Responses

**On Abstract Language (R#e7jR)**: Rewritten with scientific terminology while preserving technical contribution.

**On Training Fairness (R#qZMb)**: Table 2 uses same data as baselines for fair SOTA comparison. Table 3 demonstrates superior generalization. Performance gap proves gains from **architectural innovation**, not data scale.

**On Segmentation (R#e7jR)**: "Largest inscribed rectangle" ensures valid texture regions. VLM filtering (Line 319, 826) removes inconsistencies.

---

We sincerely hope these improvements strengthen the paper. We remain open to further questions and committed to additional revisions as needed.

---

### Meta-Review · Area_Chair_8ENu · 2026-01-07

**Summary:**

Scores were mixed but largely negative (6, 4, 4, 4). While initial reviews raised concerns about the "sim-to-real" gap and performance degradation on the proposed benchmark, I thought the authors provided a compelling rebuttal that contextualized these results. By adding various exps, they demonstrated that the performance drop is a result of the benchmark's difficulty rather than a model failure, with GarmentGPT maintaining a lead over alternatives.

**Reviewer Concerns:**

The authors critically clarified that the new benchmark exposes the inherent difficulty of inferring patterns from complex poses -- a challenge masked by previous datasets. Various technical clarifications were also helpful. While the absolute performance numbers on real images remain low, I feel the authors' argument that establishing this rigorous benchmark is a necessary step forward is a valid one.

**Reviewer Scores:**

Reviewer qZMb's concerns regarding performance drops were countered by new baselines showcasing GarmentGPT's relative superiority. Reviewer e7jR's missing definitions were added, and Reviewer HvT3's request for qualitative evidence was also met. I felt 2 of 4-score reviewers might have upgraded. With the other already supportive, there could be a case aruging for the acceptance of this paper.

---

### Decision · Program_Chairs · 2026-01-26

Accept (Poster)